# An interbacterial DNA deaminase toxin directly mutagenizes surviving target populations

Marcos H de Moraes[1], FoSheng Hsu[1], Dean Huang[2], Dustin E Bosch[3], Jun Zeng[1], Matthew C Radey[1], Noah Simon[4], Hannah E Ledvina[1], Jacob P Frick[1], Paul A Wiggins[2], S Brook Peterson[1], Joseph D Mougous[1,5,6]*

[1]Department of Microbiology, University of Washington School of Medicine, Seattle, United States; [2]Department of Physics, University of Washington, Seattle, United States; [3]Department of Laboratory Medicine and Pathology, University of Washington School of Medicine, Seattle, United States; [4]Department of Biostatistics, University of Washington School of Public Health, Seattle, United States; [5]Department of Biochemistry, University of Washington School of Medicine, Seattle, United States; [6]Howard Hughes Medical Institute, University of Washington, Seattle, United States

**Abstract** When bacterial cells come in contact, antagonism mediated by the delivery of toxins frequently ensues. The potential for such encounters to have long-term beneficial consequences in recipient cells has not been investigated. Here, we examined the effects of intoxication by DddA, a cytosine deaminase delivered via the type VI secretion system (T6SS) of *Burkholderia cenocepacia*. Despite its killing potential, we observed that several bacterial species resist DddA and instead accumulate mutations. These mutations can lead to the acquisition of antibiotic resistance, indicating that even in the absence of killing, interbacterial antagonism can have profound consequences on target populations. Investigation of additional toxins from the deaminase superfamily revealed that mutagenic activity is a common feature of these proteins, including a representative we show targets single-stranded DNA and displays a markedly divergent structure. Our findings suggest that a surprising consequence of antagonistic interactions between bacteria could be the promotion of adaptation via the action of directly mutagenic toxins.

*For correspondence: mougous@uw.edu

Competing interests: The authors declare that no competing interests exist.

## Introduction

Pathways for the delivery of toxins into contacting cells are widespread in bacteria. These include the type IV-VI secretion systems (T4-T6SS) in Gram-negative bacteria, the Esx secretion system of Gram-positives, and a number of specialized mechanisms that display a more limited distribution (*Klein et al., 2020*; *García-Bayona et al., 2017*; *Jamet et al., 2015*; *Koskiniemi et al., 2013*; *Vassallo et al., 2017*). Although most interbacterial toxins promote the competitiveness of producing organisms, the precise impact that they have on recipient cells can vary considerably. For instance, toxins that degrade the cell wall through amidase or muramidase activity lead to cellular lysis, whereas others such as nucleases cause cell death without the release of cellular contents (*Jana et al., 2019*; *Ma et al., 2014*; *Russell et al., 2011*; *Whitney et al., 2013*). Yet others, including NAD+ glycohydrolases and small ion-selective pore-forming toxins, cause growth arrest without killing (*LaCourse et al., 2018*; *Whitney et al., 2015*; *Mariano et al., 2019*). Beyond these outcomes that are detrimental to recipient cells, there are also scenarios in which toxin delivery could provide a transient benefit. Within populations of toxin-producing strains, self-intoxication is prevented through the production of specific immunity determinants that neutralize individual toxins

(*Hernandez et al., 2020*). Toxins inactivated by immunity proteins in this manner are generally assumed to have no or little impact. However, in *Burkholderia thailandensis,* it was demonstrated that a toxin delivered by the contact-dependent inhibition pathway (CDI) causes increased expression of more than 30 genes in resistant cells, including those encoding exopolysaccharide biosynthesis proteins and a T6SS (*Garcia et al., 2016*). This is associated with an increase in cellular aggregation and colony morphology changes. A separate study found that within a population of CDI toxin-producing bacteria, some cells fail to completely neutralize incoming CDI toxins delivered by their neighbors, leading to induction of the stringent response, and ultimately to enhanced antibiotic tolerance via growth arrest of this sub-population (*Ghosh et al., 2018*). The above examples highlight how interbacterial toxins can have short-term beneficial effects. Whether toxin delivery can impact the long-term evolutionary landscape of target cell populations has not been explored.

In this study, we investigate the consequences of cellular intoxication by an interbacterial toxin that acts as a cytosine deaminase. Proteins that catalyze the deamination of bases in nucleotides and nucleic acids are found in all domains of life and play essential roles in a range of physiological functions (*Iyer et al., 2011*). For instance, deaminases that target free nucleotides and nucleosides contribute to cellular homeostasis of these molecules and can be involved in the biosynthesis of modified nucleic-acid-derived secondary metabolites (*Kumasaka et al., 2007*; *Magalhães et al., 2008*; *Weiss, 2007*). RNA-targeting deaminases include the tRNA adenosine deaminase family of proteins (TAD) that contribute to tRNA maturation (*Wolf et al., 2002*), the double-stranded RNA-specific adenosine deaminase (ADAR) enzymes that affect gene regulation through the editing of mRNA and small non-coding RNA targets (*Nishikura, 2010*), and mRNA-editing cytosine deaminases APOBEC1 and members of the DYW family (*Blanc and Davidson, 2010*; *Hayes and Santibanez, 2017*). Activation-induced cytidine deaminase (AID) and APOBEC3 both target cytosine residues in single-stranded DNA and contribute to the generation of antibody diversity or control of retroviral or other retroelement replication, respectively (*Harris and Dudley, 2015*; *Feng et al., 2020*).

Bioinformatic analyses of the origins of the deaminase fold led to the surprising finding that a large and diverse collection of predicted bacterial and archaeal deaminases exhibit hallmarks of substrates of antibacterial toxin delivery pathways, including the T6SS, the Esx secretion system, and the CDI pathway (*Iyer et al., 2004*; *Makarova et al., 2019*). We recently demonstrated that one of these proteins, a substrate of an interbacterial toxin-delivering T6SS of *Burkholderia cenocepacia*, acts as a double-stranded DNA-targeting cytosine deaminase (*Mok et al., 2020*). This unusual activity stands in contrast with all previously characterized DNA-targeting deaminases, which act preferentially on single-stranded substrates. We also showed that unlike the housekeeping deaminases APOBEC3G, TadA, and Cdd, this protein, which we named DddA (double-stranded DNA deaminase A), is highly toxic when expressed heterologously in *Escherichia coli*. The mechanism by which DddA intoxicates cells was not determined.

Here, we report the discovery that DddA is a potent, direct mutagen of otherwise resistant target bacterial populations. Furthermore, we find that despite considerable differences in sequence, structure, and preferred substrates, deaminase toxins representing other subfamilies similarly possess mutagenic capacity. These results expand the range of outcomes which can result from interbacterial interactions and suggest that deaminases could play a significant role in generating genetic diversity in bacterial populations.

## Results

### DddA mediates chromosome degradation and DNA replication arrest

We previously demonstrated that DddA is a *B. cenocepacia* H111 T6SS-1 substrate that deaminates cytosine in double-stranded DNA (*Mok et al., 2020*). Cytosine deamination generates uracil, which is removed from DNA by the base excision repair (BER) pathway (*Wallace, 2014*). This process generates abasic sites and previous reports indicate that the presence of these lesions in close proximity on opposite strands can lead to double-strand DNA breaks (*D'souza and Harrison, 2003*). Thus, to begin dissecting the mechanism by which DddA leads to killing, we examined the impact of the toxin expression on chromosome stability in *E. coli*. DAPI staining coupled with fluorescence microscopy revealed that the nucleoids of cells exposed to DddA rapidly disintegrate (*Figure 1A and B*,

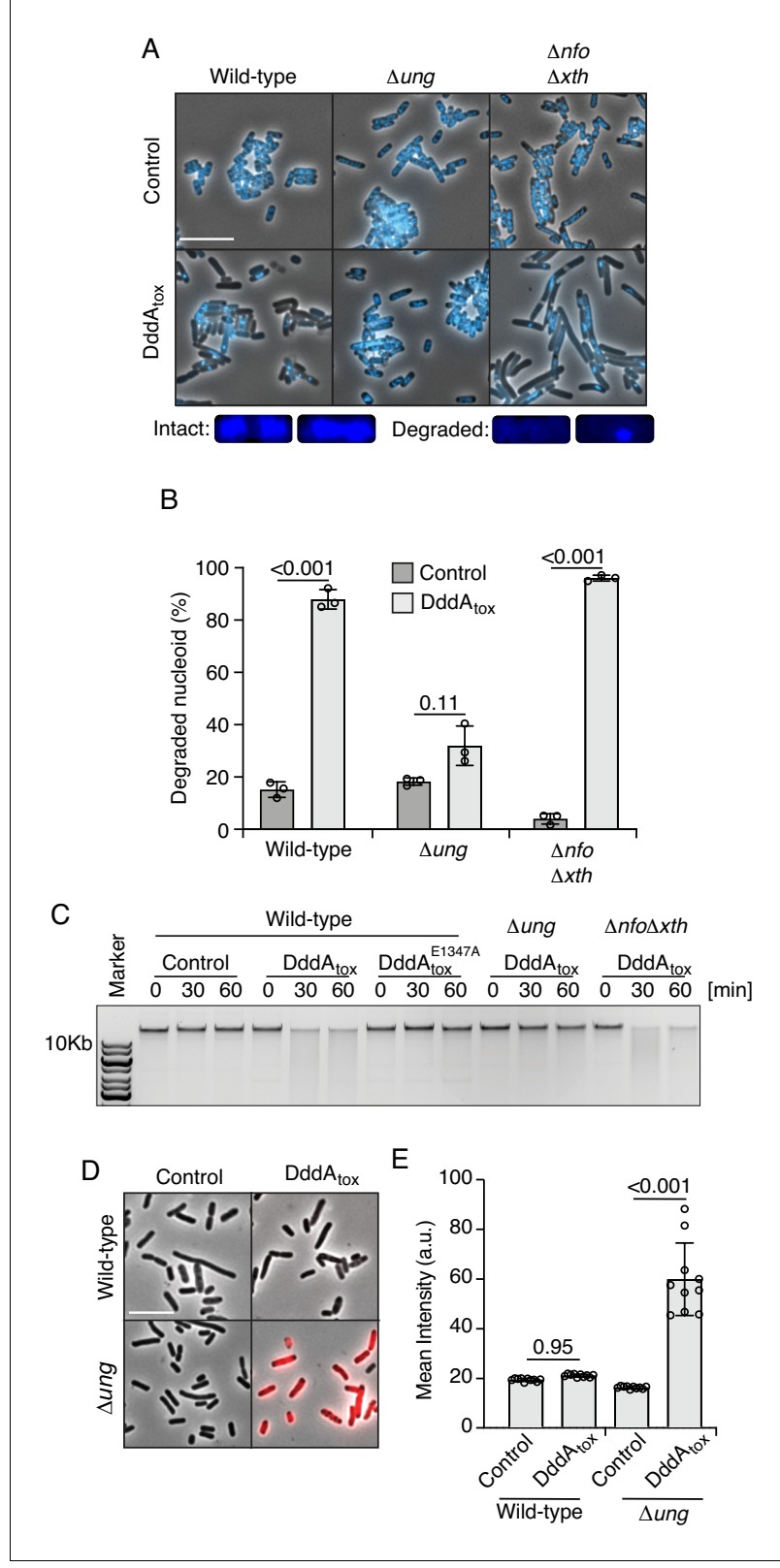

**Figure 1.** Double-stranded DNA deaminase A (DddA) expression leads to nucleoid degradation in *E. coli* wild-type cells and uracil accumulation in *E. coli* Δ*ung*. (**A**) Fluorescence microscopy of the indicated *E. coli* strains expressing DddA$_{tox}$ or carrying an empty vector (Control). DAPI staining (DNA) is shown in cyan. Top: Representative micrographs for each condition, scale bar = 10 μm. Bottom: representative images of cells with

*Figure 1 continued on next page*

*Figure 1 continued*

intact or degraded nucleoids. (**B**) Quantification of nucleoid state in cells shown in A (n ≈ 100–200 cells per condition). (**C**) Agarose gel electrophoresis analysis of total genomic DNA isolated from the indicated *E. coli* strains expressing DddA$_{tox}$, DddA$_{tox}^{E1347A}$, or carrying an empty vector (Control) after induction for the time period shown. (**D**) Fluorescence microscopy indicating genomic uracil incorporation (red) of *E. coli* strains expressing DddA$_{tox}$ or carrying an empty vector (Control), scale bar = 10 μm. (**E**) Quantification of uracil labeling signal from cells shown in D (n ≈ 50 cells per condition). Values and error bars reflect mean ± s.d. of n = 2 independent biological replicates. p-Values derive from unpaired two-tailed t-test.

The online version of this article includes the following figure supplement(s) for figure 1:

**Figure supplement 1.** Expression of DddA$_{tox}$ leads to nucleoid degradation in *E. coli.*
**Figure supplement 2.** DddA$_{tox}$ induction leads to genomic uracil accumulation in *E. coli.*

*Figure 1—figure supplement 1*). We further observed this phenomenon as genome fragmentation detectable by gel electrophoresis analysis of DNA extracted from cultures undergoing DddA-mediated intoxication (*Figure 1C*).

Uracil DNA glycosylase (Ung) initiates BER by removing uracil (*Yonekura et al., 2009*). We therefore investigated how cells lacking Ung activity (Δ*ung*) respond to DddA expression. The level of uracil in genomic DNA can be measured by treating fixed cells with a fluorescently conjugated, catalytically inactive Ung protein (*Róna et al., 2016*). Employing this tool, we found that, as predicted, Ung inactivation leads to the accumulation of uracil in the DNA of DddA-intoxicated cells (*Figure 1D and E*, *Figure 1—figure supplement 2*). Additionally, Ung inactivation alleviated both nucleoid disruption and DNA fragmentation in *E. coli* populations expressing DddA (*Figure 1A–C*). In contrast, deletions of genes encoding the endonucleases Xth and Nfo, which operate downstream of Ung in BER by removing DNA abasic sites, had no impact on DddA-induced nucleoid disintegration. These findings collectively link the effects of DddA on chromosome integrity to uracil removal from DNA by the BER pathway.

If Ung-catalyzed removal of uracil from DNA and subsequent chromosome fragmentation is responsible for DddA-mediated killing, we reasoned that Ung inactivation should affect susceptibility to DddA. However, we found that *E. coli* wild-type and Δ*ung* were equally susceptible to intoxication (*Figure 2A*). It was previously demonstrated that starvation for thymine, which leads to an increase of uracil incorporation in DNA, can disrupt DNA replication complexes, killing cells in a process known as thymineless death (TLD) (*Khodursky et al., 2015*). Accordingly, we investigated whether DNA replication is affected by DddA. In both wild-type and Δ*ung E. coli* strains, DddA induction resulted in the rapid loss of fluorescent foci formed by YPet-labeled DnaN, an established indicator of replication fork collapse (*Aakre et al., 2013*; *Reyes-Lamothe et al., 2010*; *Figure 2B–D*, *Figure 2—figure supplement 1A and B*, *Video 1* and *Video 2*).

To further probe how DddA-intoxicated cells die, we sequenced the transcriptome of *E. coli* cells expressing DddA for 1 hr – a time point by which > 99% of cells are no longer viable (*Figure 2A*). Despite substantial sequencing depth, and the known ability of RNA polymerase to readily incorporate adenine opposite uracil residues encountered in DNA (*Brégeon et al., 2003*; *Viswanathan and Doetsch, 1998*), this experiment yielded no evidence for the incorporation of mutations into transcripts (*Figure 2—figure supplement 2A and B*). However, genomic DNA sequencing at this time point revealed widespread C•G-to-T•A transitions in the preferred context for DddA (5′-TC-3′), consistent with the uracil enrichment we observed by fluorescent labeling (*Figure 1D and E*, and *Figure 2E–G*). We note that genomic DNA sequencing could only be conducted on the Δ*ung* background, as nucleoid deterioration of intoxicated wild-type cells prohibited sequencing library construction (*Figure 1A–C*). However, we previously found that repeated exposure to a low level of DddA expression leads to accumulation of C•G-to-T•A mutations in the wild-type strain (*Mok et al., 2020*). Taken together with our nucleoid integrity and DNA replication reporter data, these findings suggest that the fate of cells intoxicated by DddA is determined prior to nucleoid deterioration and prior to or coincident with the inhibition of transcription. Our findings do not rule out a mechanistic overlap between DddA- and TLD-mediated cell killing. In this regard, it is noteworthy that despite a considerable volume of research spanning several decades, the molecular underpinnings of TLD remain incompletely understood (*Khodursky et al., 2015*).

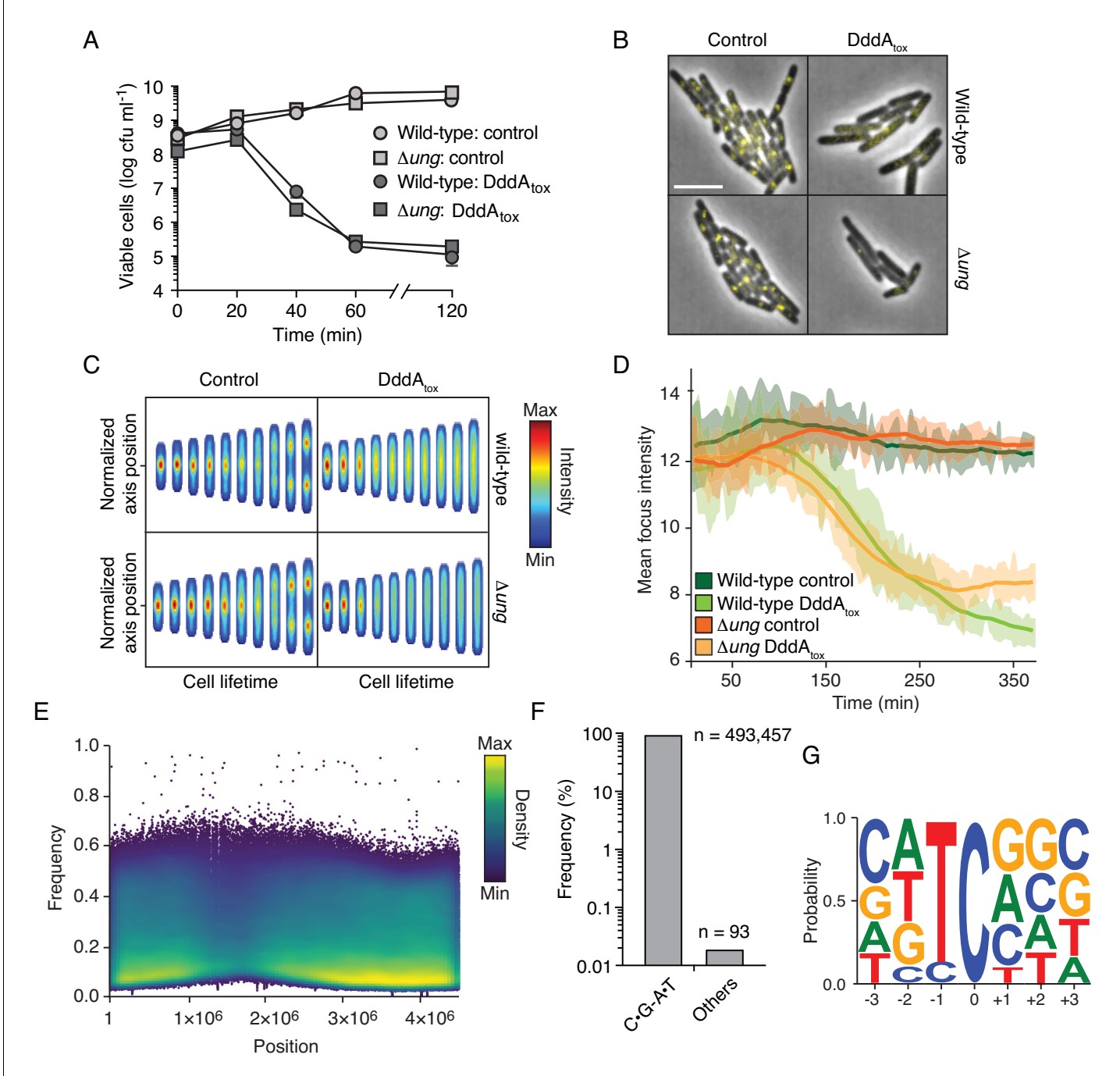

**Figure 2.** Intoxication by DddA leads to DNA replication arrest and widespread uracil incorporation across the genome. (A) Viable cells (colony-forming units, cfu) of the indicated *E. coli* strains recovered following induction of $DddA_{tox}$ or empty vector (Control) for the time period shown. Values represent mean ± s.d. of n = 3 technical replicate and data are representative of three independent experiments. (B) Representative images from time-lapse microscopy of DnaN-YPet-expressing strains 6 hr post-induction of $DddA_{tox}$, scale bar = 5 μm. (C) Cell tower representation of averaged localized fluorescence intensity of DnaN-YPet-expressing strains shown in B over the course of cell lifetimes. (n ≈ 20–300 cells per condition at start of experiment). (D) Mean focus intensity of each frame during time-lapse microscopy of DnaN-YPet-expressing strains over the course of 6 hr (n ≈ 20–300 cells per condition at start of experiment). (E) Representation of single-nucleotide variants (SNVs) by chromosomal position, frequency, and density in *E. coli Δung* following 60 min expression of $DddA_{tox}$. (F) Frequency of the indicated substitutions among the SNVs shown in E. (G) Probability sequence logo of the region flanking mutated cytosines among the SNVs shown in (E).

The online version of this article includes the following figure supplement(s) for figure 2:

**Figure supplement 1.** Expression of $DddA_{tox}$ in *E. coli* leads to replication arrest but does not mutagenize RNA.

*Figure 2 continued on next page*

Figure 2 continued

**Figure supplement 2.** Expression of DddA_tox does not mutagenize RNA.

## DddA intoxication consequences vary among recipient species

The experiments described above were performed in *E. coli* heterologously expressing DddA, which may not capture the physiological impact of the toxin on recipient cell populations when it is delivered by the T6SS of *B. cenocepacia*. As a first step toward assessing the effect of DddA on recipient cells, we performed interbacterial competition assays between *B. cenocepacia* wild type or a strain bearing catalytically inactive DddA (*dddA*^E1347A) and *E. coli* (**Figure 3A**). Surprisingly, we found no evidence of DddA-mediated inhibition of *E. coli* in these experiments. A straightforward explanation for these results is that T6SS-1 of *B. cenocepacia* is unable to deliver toxins to *E. coli*. However, we observed that a *B. cenocepacia* strain lacking T6SS-1 activity (Δ*icmF*) exhibits reduced competitiveness toward *E. coli*, indicating that toxin delivery can occur between these organisms.

Our finding that DddA delivered to *E. coli* via the T6SS of *B. cenocepacia* fails to result in killing led us to question whether the toxin generally exhibits this property when delivered to other species or whether it can be lethal against select recipients beyond a *B. cenocepacia* strain sensitized to intoxication via inactivation of the DddA immunity determinant DddA_I (**Mok et al., 2020**). To evaluate this, we performed additional competition experiments between *B. cenocepacia* WT or *dddA*^E1347A and a panel of Gram-negative organisms (**Figure 3A–D**). We found that the effect of DddA delivery via the T6SS varied; in general, enteric species and *Acinetobacter baumannii* (**Figure 3A and B**) resisted intoxication, whereas other species, including *Pseudomonas aeruginosa*, *P. putida*, and other *Burkholderia* species (**Figure 3C and D**), were highly sensitive to DddA-mediated killing. Given that *dddA* is encoded in only 97 of 308 *B. cenocepacia* strains with publicly available genomes and is missing from widely employed model strains of this species (J2315 and K56-2), we also evaluated whether it could mediate intoxication of a strain lacking the effector gene and its accompanying immunity determinant, *dddA_I*. We found that this strain, *B. cenocepacia* K56-2, was susceptible to intoxication to a level similar to other *Burkholderia* species tested (**Figure 3D**). Notably, in all the sensitive bacteria, DddA accounted for all the inhibitory activity of the T6SS of *B. cenocepacia* under the conditions of our assay.

Given that experiments we initially performed to investigate DddA mechanism employed a species resistant to intoxication (*E. coli*) by physiological levels of the toxin, we next sought to determine whether killing of a species susceptible to DddA delivered intercellularly shared similar mechanistic hallmarks. First, we demonstrated by mass spectrometry and cellular fluorescence assays that the sensitive species *P. aeruginosa* accumulates uracil in its DNA when grown in contact with wild-type *B. cenocepcia,* but not a mutant bearing catalytically inactive DddA (**Figure 3E**, **Figure 3—figure supplement 1**). Moreover, the detection of DddA-mediated uracil accumulation in these assays required inactivation of *ung*, as observed during heterologous expression of DddA in *E. coli*. Additionally, single-nucleotide variant (SNV) analysis of genomic DNA sequences from *P. aeruginosa* Δ*ung* grown in contact with *B. cenocepacia* revealed the accumulation of C to T mutations, consistent with DddA delivered via T6SS acting in a similar fashion to the heterologously expressed protein (**Figure 3F** and **Figure 3—figure supplement 2**). Finally, we confirmed that ectopic expression of DddA in *P. aeruginosa* leads to a similar pattern of nucleoid deterioration as that seen in *E. coli* (**Figure 3—figure supplement 3**). Interestingly, the susceptibility of *P. aeruginosa* to DddA intoxication was not influenced by *ung* inactivation (**Figure 3G**), consistent with our earlier conclusion that cell death mediated by the toxin occurs upstream of massive uracil accumulation and widespread chromosome fragmentation (**Figure 1**).

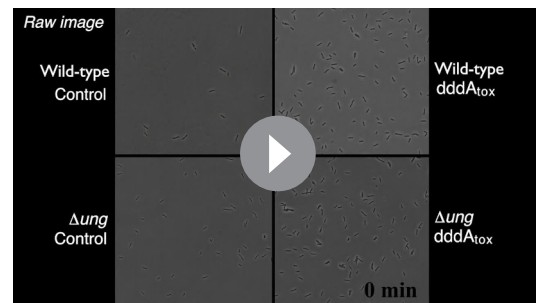

**Video 1.** Double-stranded DNA deaminase A (DddA) inhibits *E. coli* growth. Time lapse phase microscopy of *E. coli* expressing DddA during a 300-min period.
https://elifesciences.org/articles/62967#video1

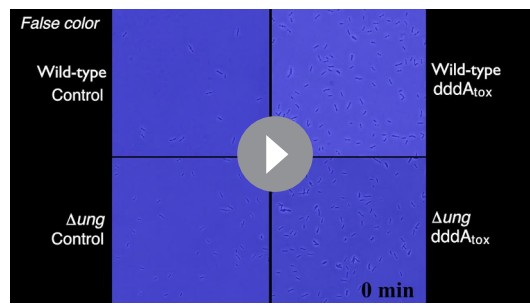

**Video 2.** Double-stranded DNA deaminase A (DddA) arrests replication in *E. coli* growth. Time lapse fluorescence microscopy of *E. coli* with YPet-labeled DnaN-expressing DddA during a 300-min period.
https://elifesciences.org/articles/62967#video2

## Bacteria resistant to DddA-mediated intoxication accumulate DddA-catalyzed mutations

Since our earlier results clearly demonstrated the capacity of DddA to act within *E. coli* – a species resistant to DddA-mediated intoxication during interbacterial competition – we next tested whether sub-lethal DddA activity in resistant bacterial could lead to the accumulation of mutations. Precedence for the mutagenic activity of cytosine deaminases at physiological levels is well established for eukaryotic enzymes, including those of the APOBEC and AID families (*Burns et al., 2013*; *Maul and Gearhart, 2010*). Strikingly, we found that *E. coli* populations subject to intoxication by T6SS-1 of wild-type *B. cenocepacia* for a single hour display a nearly 10-fold increase in the frequency of rifampicin-resistant (Rif[R]) cells compared to those exposed to *B. cenocepacia dddA*[E1347A] (*Figure 4A*). We next used DNA sequencing to establish whether these mutations derive directly from the activity of DddA. Resistance to rifampicin can result from at least 69 base substitutions distributed over 24 positions in *rpoB*, which encodes the β-subunit of RNA polymerase (*Garibyan et al., 2003*). Sequencing of *rpoB* from 25 independent clones obtained from *E. coli* populations grown in competition with *B. cenocepacia* revealed that in all but one of these, rifampicin resistance was conferred by C•G-to-T•A transitions within the preferred context of DddA (*Figure 4B*, *Supplementary file 1*). In contrast, a study of 152 spontaneous Rif[R] mutants in *E. coli* found only 32 such mutations in *rpoB* (*Garibyan et al., 2003*), indicating significant enrichment for this sequence signature in clones deriving from DddA-intoxicated populations (Fisher's exact test, p<0.001). Whole-genome sequencing (WGS) of 10 Rif[R] clones isolated from these experiments led to the identification of an additional 12 mutations outside of *rpoB* distributed across these strains, and all were C•G-to-T•A transitions in a 5′-TC-3′ context (*Figure 4B and C*, *Supplementary file 1*). Both a Fisher exact test and a negative binomial regression analysis indicated a significant enrichment for C•G-to-T•A mutations in the context preferred by DddA in intoxicated populations when compared to the pattern of mutations found to arise spontaneously in *E. coli* under neutral selection (p<0.005) (*Lee et al., 2012*). Together, these data provide compelling evidence that DddA delivered to *E. coli* during interbacterial competition drives mutagenesis.

We next examined whether bacteria resistant to DddA-mediated killing generally exhibit an accumulation of mutations following intoxication. Among the resistant organisms, *Klebsiella pneumoniae* and enterohemorrhagic *E. coli* (EHEC) displayed higher Rif[R] frequency when in contact with *B. cenocepacia* containing active DddA than when in contact with *B. cenocepacia* bearing the *dddA*[E1347A] allele (*Figure 4D and E*); *Salmonella enterica* and *Acinetobacter baumanii* did not exhibit this behavior (*Figure 4—figure supplement 1A and B*). Similar to our findings in *E. coli*, *rpoB* genes from Rif[R] *K. pneumoniae* and EHEC clones isolated from cells exposed to wild-type *B. cenocepacia* exhibited mutagenesis signatures consistent with DddA activity (*Figure 4E,F,H and I*, *Supplementary file 1*). We further probed the mutagenic potential of DddA within these clones by performing WGS. Within the 11 *K. pneumoniae* isolates we sequenced, 50 of the 51 SNPs detected were C•G-to-T•A transitions, and each of these was located in the 5′-TC-3′ context preferred by DddA (*Mok et al., 2020*), a highly significant enrichment compared to spontaneous mutation patterns (p<0.001) (*Lee et al., 2012*). The EHEC isolates we sequenced shared this trend, with 82 of 88 SNPs across seven strains corresponding to C•G-to-T•A transitions in the context preferred by DddA. We found no indication of clustering of DddA-induced mutations across genomes of *E. coli*, *K. pneumoniae*, or EHEC, suggesting that mutagenesis by the toxin occurs in a random fashion (*Figure 4C,F and I*).

Finally, we asked whether species susceptible to killing by DddA could also accumulate mutations in cells that survive intoxication. We observed no evidence of increased mutation frequency in populations of the susceptible competitors *B. cenocepacia* K56-2, *B. ambifaria*, and *P. aeruginosa* (*Figure 4—figure supplement 1C–E*). Furthermore, the genomes of 65 Rif[R] *P. aeruginosa* clones

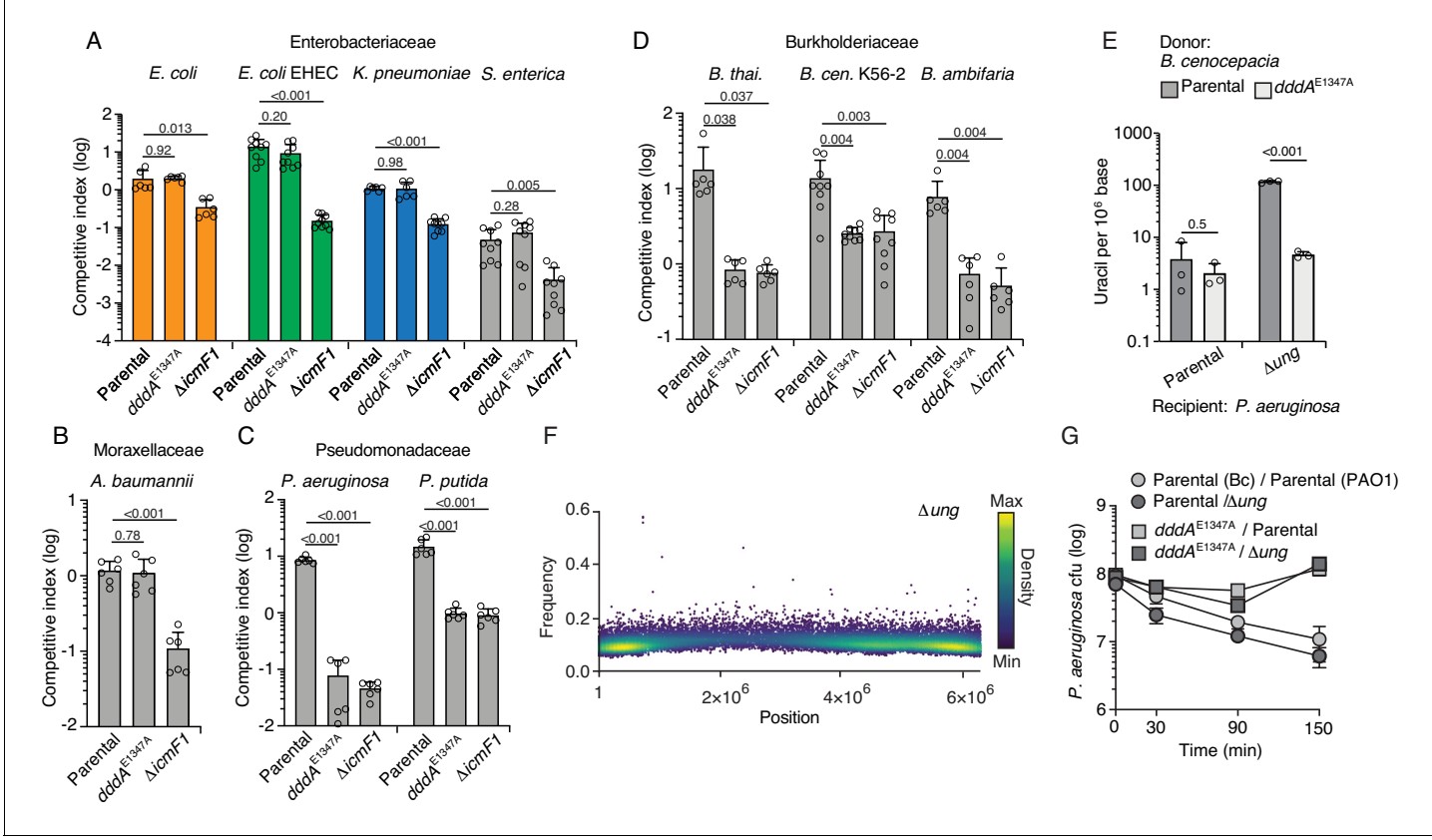

**Figure 3.** Double-stranded DNA deaminase A (DddA) delivery via the T6SS during interbacterial competition results in varying outcomes among different recipient species. (A–D) Competitiveness of the *B. cenocepcia* strains indicated at bottom against selected Enterobacteriaceae (colors relate to *Figure 4*) (A), Moraxellaceae (B), Pseudomonadaceae (C), or Burkholderiaceae (D). Pairs of organisms were cocultured on a solid surface for 6 hr. (E) Mass spectrometric quantification of uracil in genomic DNA obtained from 1 hr co-cultures of the indicated strains of *P. aeruginosa* with *B. cenocepacia* wild-type or *dddA*$^{E1347A}$. Values and error bars reflect mean ± s.d. of n = 3 independent biological replicates. p-Values derive from unpaired two-tailed t-test. (F) Single-nucleotide variants (SNVs) detected in populations of *P. aeruginosa* Δ*ung* after 1 hr growth in competition with *B. cenocepacia*. SNVs are plotted according to their chromosomal position and frequency and colored according to their relative density. (G) Cell viability of the indicated *P. aeruginosa* strains after 1 hr growth in competition with *B. cenocepacia* wild-type or *dddA*$^{E1347A}$. Values and error bars reflect mean ± s.d. of n = 2 independent biological replicates.

The online version of this article includes the following figure supplement(s) for figure 3:

**Figure supplement 1.** *P. aeruginosa* accumulates genomic uracil during competition with *B. cenocepacia*.

**Figure supplement 2.** Delivery of double-stranded DNA deaminase A (DddA) to *P. aeruginosa* Δ*ung* during interbacteria competition with *B. cenocepacia* results in the accumulation of C•G-to-T•A transitions.

**Figure supplement 3.** Expression of DddA$_{tox}$ in *P. aeruginosa* results in nucleoid degradation.

derived from interbacterial growth competition experiments between *P. aeruginosa* and *B. cenocepacia* revealed mutations exclusively within *rpoB*, and only 15% of these mutations conformed to the mutagenic signature of DddA (*Supplementary file 1*). However, intoxication of *P. aeruginosa* Δ*ung* by DddA yielded elevated mutation frequency, resembling that of Enterobacteriaceae exposed to the toxin (*Figure 4—figure supplement 1F*). These results suggest that mutagenesis is likely a rare outcome of DddA-mediated intoxication in sensitive competitor species, but raise the possibility that it may become more frequent in strains unable to repair DddA-catalyzed lesions in DNA.

## Diverse deaminase toxins have mutagenic activity

Our discovery that DddA can be a mutagen in bacterial communities prompted us to investigate whether deaminase toxins might more broadly impact the mutational landscape of bacteria. Aravind and colleagues describe 22 sequence divergent subfamilies into which enzymes belonging to the deaminase superfamily distribute (*Iyer et al., 2011*). Among these, eight subfamilies possess

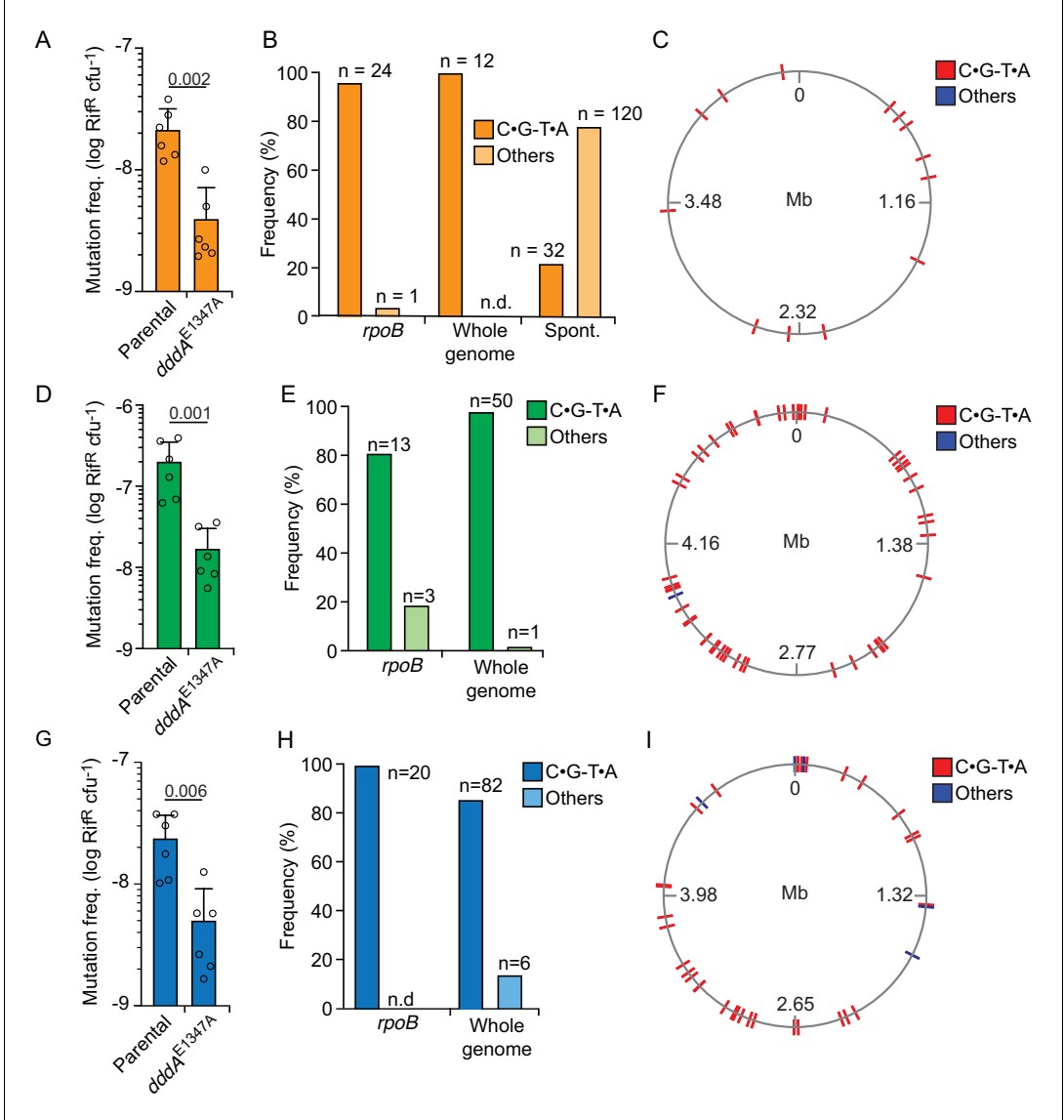

**Figure 4.** Delivery of double-stranded DNA deaminase A (DddA) by *B. cenocepacia* induces mutagenesis in a subset of resistant recipient species. (**A**, **D**, and **G**) Mutation frequency as measured by spontaneous rifampicin resistance frequency in clones of different species recovered from growth in competition with *B. cenocepacia* wild-type or *B. cenocepacia dddA*$^{E1347A}$ (n = 2). (**B**, **E**, and **H**). Distribution of different mutation types detected in *rpoB* or the whole genome of rifampicin-resistant clones of different species recovered after growth in competition with wild-type *B. cenocepacia*. (**B**) Pattern of spontaneous mutation types observed in *rpoB* of *E.coli* derived from *Garibyan et al., 2003*. (**C**, **F**, and **I**). Genome distribution of different mutation types detected by whole-genome sequencing of rifampicin-resistant clones recovered after competition with *B. cenocepacia*. Species targeted include *E. coli* (**A–C**), *E. coli* EHEC (**D–F**), and *K. pneumoniae* (**G–I**). Colors relate to *Figure 3*. Values and error bars reflect mean ± s.d. of n = 2 independent biological replicates with n = 3 technical replicates each. p-values derive from unpaired two-tailed t-test. Spont. Spontaneous.

The online version of this article includes the following figure supplement(s) for figure 4:

**Figure supplement 1.** Double-stranded DNA deaminase A (DddA)-mediated intoxication is not mutagenic during competition between *B. cenocepacia* and certain recipient species.

members associated with interbacterial toxin systems (*Figure 5A*). For instance, DddA belongs to subfamily SCP1.201-like, herein renamed *Bacterial Deaminase Toxin Family 1* (BaDTF1), composed of members from Proteobacteria and Actinobacteria. We selected representatives from two additional deaminase superfamily subgroups containing predicted interbacterial toxins: *P. syringae* WP_011168804.1 of the DYW-like subgroup, of which we reassigned the bacterial toxin members to BaDTF2 and *Taylorella equigenitalis* WP_044956253.1 of the Pput_2613-like subgroup, renamed

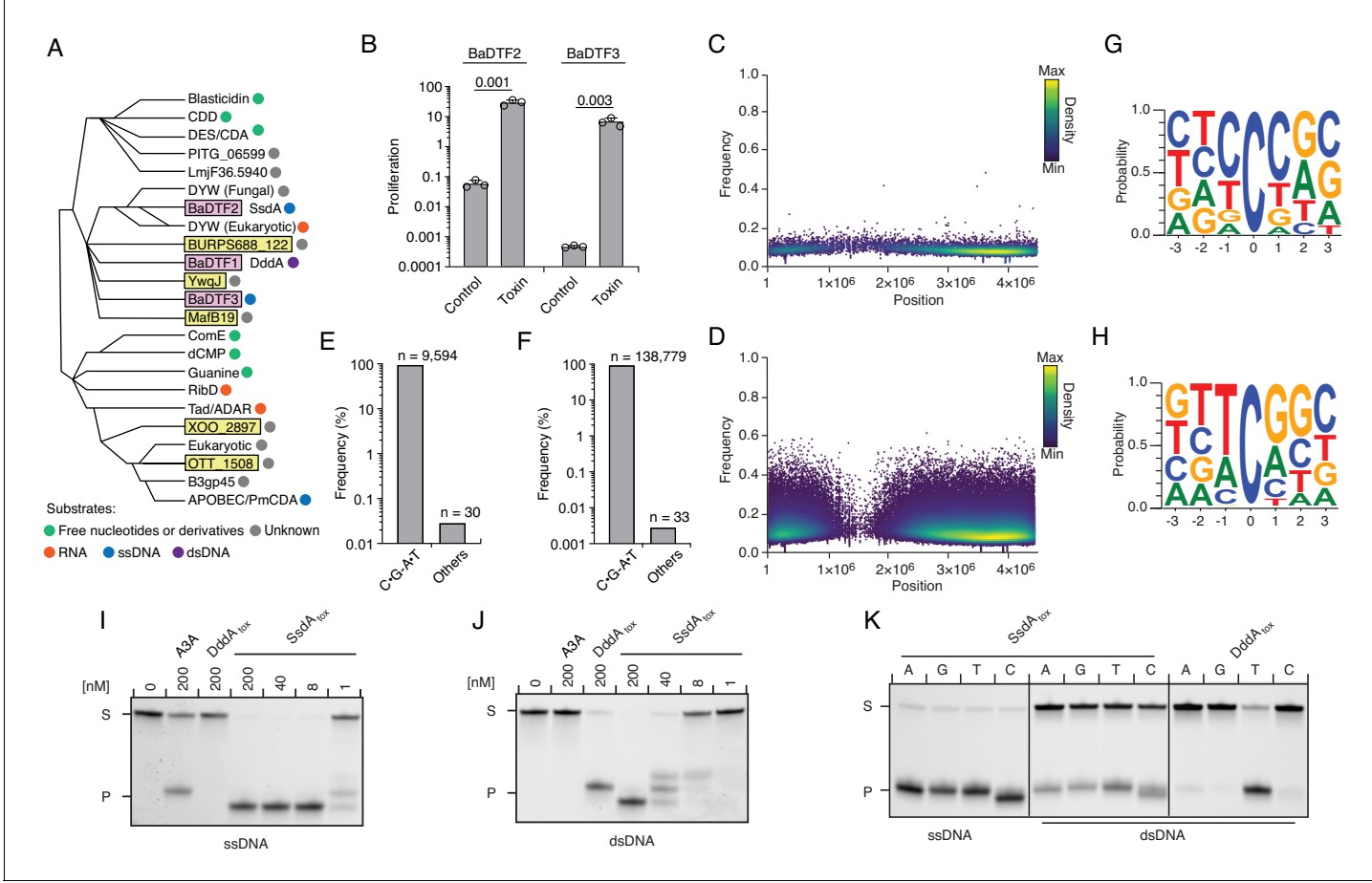

**Figure 5.** Predicted deaminase toxins from BaDTF2 and BaDTF3 clades exhibit mutagenic activity and a BaDTF2 representative targets ssDNA. (A) Dendogram indicated evolutionary history of clades within the deaminase superfamily their predicted substrates (colored dots), modified from *Iyer et al., 2011*. Predicted toxins with unknown substrates, yellow boxes; deaminases toxins with defined biochemical activity, pink boxes. (B) Toxicity of representative BaDTF2 and BaDTF3 toxins as indicated by the proliferation (fold change in colony-forming unit [cfu] recovered) of *E. coli* after 1 hr expressing the toxins or the empty vector (Control). Values represent means ± s.d., and p-values derive from unpaired two-tailed t-test. (C and D) Representation of single-nucleotide variants (SNVs) by chromosomal position, frequency, and density in *E. coli* Δ*ung* after 1 hr expression of representative BaDTF2 (C) or BaDTF3 (D) toxins. (E and F). Distribution of different nucleotides substitutions among SNVs detected in *E. coli* Δ*ung* expressing representative BaDTF2 (E) or BaDTF3 (F) toxins. (G and H) Probability sequence logo of the region flanking mutated cytosines from *E. coli* Δ*ung* intoxicated with representative BaDTF2 (G) or BaDTF3 (H) toxins. (I and J). In vitro cytidine deamination assays for BaDTF2 toxin single-strand DNA deaminase toxin A (SsdA) using a single-stranded (I) or double-stranded (J) FAM-labeled DNA substrate 'S' with cytidines in the contexts CC, TC, AC, and GC. Cytidine deamination leads to products 'P' with increased mobility. A3A, APOBEC3A (control for activity on ssDNA). DddA$_{tox}$ was used as a control for activity toward dsDNA. K. In vitro cytidine deamination assays for SsdA$_{tox}$ or DddA$_{tox}$ using a single-stranded or double-stranded FAM-labeled DNA substrate with a single cytidine in the context indicated at top. Gels shown in I-K are representative of two replicates. Data in B represent means ± s.d., p-values derive from unpaired two-tailed t-test (n = 3).

The online version of this article includes the following figure supplement(s) for figure 5:

**Figure supplement 1.** SsdA$_{tox}$ exhibits cytidine deaminase activity toward RNA in vitro, but not in vivo.

BaDTF3 (*Figure 5A*). Like DddA, both exhibited toxicity when heterologously expressed in *E. coli* (*Figure 5B*). We then evaluated whether these toxins have mutagenic activity by sequencing DNA extracted from *E. coli* expressing each protein. To both increase our sensitivity for detecting cytosine deaminase activity and to circumvent the potential for barriers to sequencing caused by BER-generated lesions in DNA (e.g. abasic sites and double stranded breaks), we employed the Δ*ung* background in these studies. Remarkably, despite their sequence divergence from DddA, representatives from both BaDTF2 and BaDTF3 introduced high levels of C•G-to-T•A transitions (*Figure 5C–F*). However, each operated within a unique sequence context, which differed from that preferred by DddA.

The BaDTF2 representative preferentially targeted cytosine located within pyrimidine tracks (5′-YCY), but could also deaminate cytosines in all other contexts with lower efficiency (*Figure 5G*). In contrast, the BaDTF3 member targeted cytosine most efficiently when preceded by thymidine, and less efficiently when preceded by adenosine or cytosine; it did not act on cytosines preceded by guanosine (5′-HC) (*Figure 5H*).

Previously characterized proteins in the DYW deaminase family are found in plants and protists and participate in C to U editing of specific organellar mRNA transcripts (*Hayes and Santibanez, 2017*; *Oldenkott et al., 2019*; *Shikanai, 1847*). In these organisms, the DYW deaminase domain is generally found at the C-terminus of a larger polypeptide characterized by pentatricopeptide repeats (PPRs), which recruit the catalytic domain of the protein to target mRNAs (*Barkan and Small, 2014*). The mutagenic activity of *P. syringae* WP_011168804.1 suggested that contrary to other DYW proteins, the substrate range of this BaDTF2 member could include DNA. To examine this biochemically, we expressed and purified the toxin domain of the *P. syringae* enzyme and performed in vitro deamination assays on single-stranded and double-stranded DNA templates. Unlike DddA, the BaDTF2 protein exhibited potent cytosine deaminating activity toward ssDNA (*Figure 5I*). Cytosine residues in dsDNA were also targeted, but with considerably lower efficiency (*Figure 5J*). Based on these data, we named the *P. syringae* representative of the BaDTF2 subfamily single-strand DNA deaminase toxin A (SsdA). Consistent with the in vivo mutagenesis results, purified SsdA$_{tox}$ could target cytosine residues preceded by any of the four bases (*Figure 5K*). Notably, we detected only residual activity of SsdA$_{tox}$ toward RNA targets in vitro, suggesting that DNA is the physiologically relevant substrate of the toxin (*Figure 5—figure supplement 1A*). This is supported by RNA-seq analysis of *E. coli* cells expressing SddA$_{tox,}$ which revealed no detectable increase in the number of C•G-to-T•A transitions in cDNA sequences (*Figure 5—figure supplement 1B and C*).

## The SsdA structure specifies a new group of bacterial deaminases

To begin to understand the basis for the distinct substrate preference exhibited by SsdA, we determined its structure in complex with its immunity determinant, SsdA$_I$, to 3.0 Å (*Figure 6A* and *Supplementary file 2*). The structure of SsdA exhibits the basic characteristics of deaminase enzymes, including active site histidine and cysteine residues in position to coordinate a zinc ion, and the five β-strands (S1-5) and three α-helices (H1, H3, and H4) that constitute the core fold of deaminase superfamily enzymes (*Figure 6B*; *Iyer et al., 2011*). More specifically, SsdA groups with the C-terminal hairpin division of the superfamily, and consistent with this assignment, S4 and S5 of SsdA are antiparallel. This is a notable divergence from all other characterized deaminases that act on ssDNA, for instance members of the APOBEC family, wherein these strands are parallel. Outside of its basic fold, SsdA bears little structural homology with characterized deaminases, including DddA (*Figure 6C–E*). Indeed, SsdA bears most overall structural similarity with the folate-dependent transformylase domain of PurH (DALI; Z score, 7.7), which contains the deaminase fold despite its sequence and functional divergence from deaminase enzymes (*Iyer et al., 2011*; *Wolan et al., 2002*).

SsdA$_I$ is an exclusively α-helical protein that shares a large (~1300 Å$^2$) interface with SsdA. As is often observed in toxin–immunity co-crystal structures, this interface effectively demarcates the active site of SsdA (*Figure 6A*; *Hernandez et al., 2020*). Much of the SsdA-interacting surface of SsdA$_I$ is composed of a protrusion that includes an extended loop and two short helical segments that project residues into the active site cavity of SsdA (*Figure 6F*). Interestingly, these residues contact a conserved Ser-Gly-Trp (SGW) motif that resides in close proximity to the active site of SsdA and is unique to the BaDTF2 members of the DYW subgroup.

SsdA represents the first member of proteins classified into the DYW-family to be structurally characterized, precluding a direct comparison with proteins from this family known to target mRNA. However, the fact that SsdA displays a substrate preference distinct from the related plant and protist proteins led us to search for sequence elements that differ between BaDTF2 members and canonical DYW proteins. Our search revealed several conserved features of DYW proteins previously demonstrated to be important for mitochondrial or chloroplast gene editing in vivo that are absent from the BaDTF2 proteins. These include the eponymous C-terminal DYW residues, a CxC motif believed to be important for coordinating a second $Zn^{2+}$ atom, and a proline-glycine (PG) motif located within a characteristic insertion between S2 and H2 of the deaminase fold

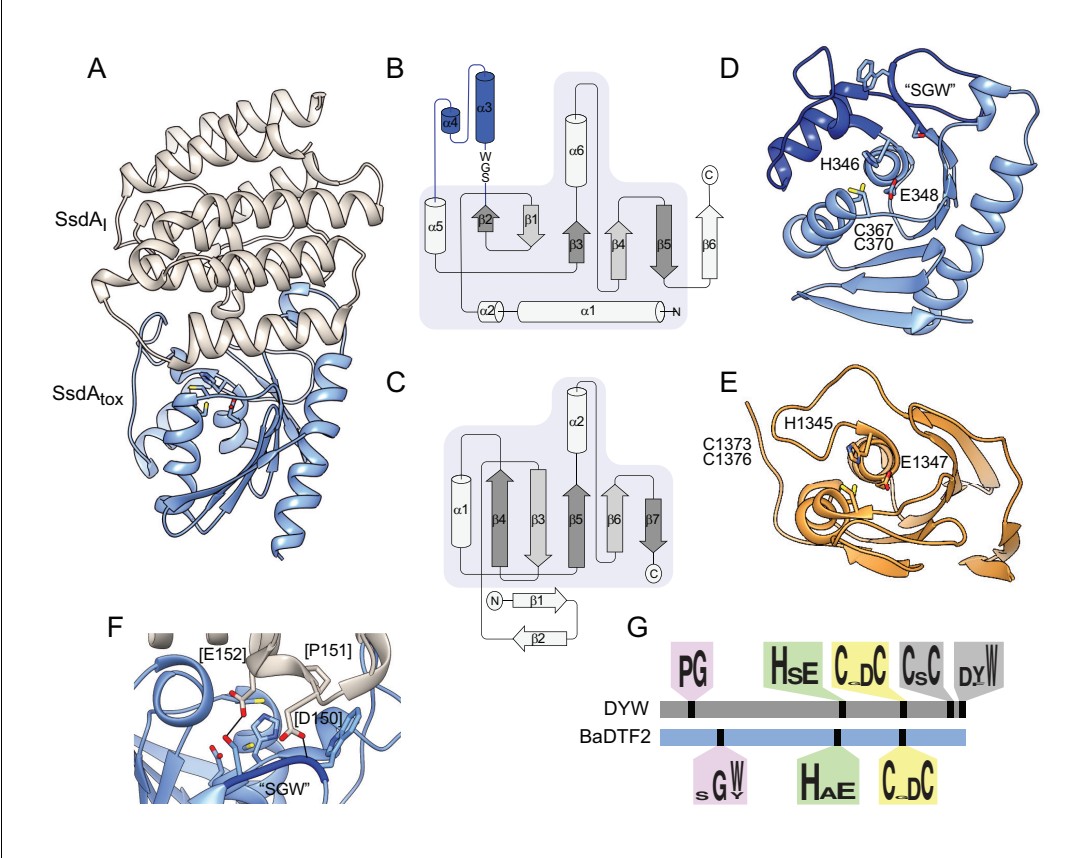

**Figure 6.** The structure of BaDF2 member single-strand DNA deaminase toxin A (SsdA) bears little resemblance to double-stranded DNA deaminase A (DddA) and reveals motifs differentiating toxins from RNA-targeting DYW proteins. (**A**) Ribbon diagram depiction of the SsdA$_{tox}$-SsdA$_I$ structure. (**B,C**) Secondary structure diagrams for SsdA$_{tox}$ and the core fold of deaminase superfamily proteins (*Iyer et al., 2011*). The SWG motif conserved in BaDTF2 toxins and the accompanying α-helical insertion (blue) are indicated in B. (**D**) Active site view of SsdA$_{tox}$. Catalytic and zinc-coordinating residues, SGW motif and α-helical insertion (blue) are indicated. (**E**) Active site view of DddA$_{tox}$ indicating catalytic and zinc-coordinating residues. (**F**) Zoom-in view of the contact site between SsdA$_I$ and the active site of SsdA$_{tox}$. (**G**) Conserved motifs identified in DYW and BaDTF2 proteins.

(*Figure 6G*; *Boussardon et al., 2014*; *Gutmann et al., 2020*; *Wagoner et al., 2015*). Instead, BaDTF2 proteins contain the aforementioned SGW motif within this insertion. Given the significant divergence of functionally critical regions between DYW and BaDTF2 proteins, we propose that BaDTF2 members constitute a new deaminase subfamily. The universal link of BaDTF2 proteins to interbacterial antagonism pathways leads us to speculate that BaDTF2 proteins beyond SsdA are likely to target ssDNA.

## Discussion

In this work, we have shown that interbacterial deaminase toxins can act as potent mutagens of target cells. This discovery provides a previously unrecognized, and potentially widespread mechanism by which bacteria can acquire genetic diversity that allows them to adapt to changing environmental conditions. This can have important phenotypic consequences; for example, we found that a deaminase toxin can increase the frequency of rifampicin resistance. The ecological ramifications of mutagenesis by deaminase toxins are not yet understood. There are myriad sources of single-nucleotide substitutions within bacterial populations. These include endogenous cellular maintenance activities such as replication and metabolism, and environmental stresses including xenobiotics and ionizing radiation (*Schroeder et al., 2018*; *Ciccia and Elledge, 2010*). Studies often report C•G-to-T•A transitions as the most common mutation type observed within evolving bacterial populations. Unfortunately, the majority of these studies are performed on monocultures of in vitro grown bacteria,

limiting their applicability to our understanding of deaminase toxins delivered between species (*Lee et al., 2012*; *Dettman et al., 2016*; *Wielgoss et al., 2011*). Nevertheless, they do provide some insights into the origins of cytosine mutations in natural contexts, as they implicate spontaneous cytosine deamination in the GC skew observed between leading and lagging strands on bacterial chromosomes (*Bhagwat et al., 2016*). Relative to spontaneous cytosine deamination, the global contribution of cytosine deaminase toxins to the landscape of cytosine mutations is likely minor. However, under certain circumstances, cytosine deaminase toxins could significantly impact evolutionary outcomes. For instance, based on the ability of DddA to install multiple mutations on a single genome during one intoxication event, deaminase toxins may facilitate the emergence of phenotypes that are otherwise slow to evolve. This could be clinically relevant to the treatment of polymicrobial infections, where bacteria that possess deaminase toxins are present and the resistance to certain antibiotics requires a succession of mutagenic events (*Cabot et al., 2014*; *van der Putten et al., 2019*).

Our study leads us to question whether mutagenesis of target cells by deaminase toxins occurs opportunistically, or whether this phenomenon could be an evolutionarily selected property of the toxin in certain instances. There is strong evidence that under certain circumstances hypermutator phenotypes proliferate within bacterial populations (*Barrick and Lenski, 2013*; *Mena et al., 2008*; *Denamur et al., 2000*). In such cells, the fitness contribution of beneficial mutations outweighs the reduction on fitness by other, deleterious mutations (*Denamur and Matic, 2006*; *Jayaraman, 2011*). In spatially structured, cooperating bacterial systems, it is conceivable that such hypermutator phenotypes could be transiently achieved with the aid of neighboring cells via the delivery of a mutagenic toxin. Additionally, our data show that the same deaminase toxin can mutagenize certain bacteria while causing death in others. Therefore, a toxin that acts as a mutagen of a cooperating bacterium could also remain as a component of the antibacterial arsenal of the producer.

With a relatively small sampling of bacterial diversity, we found both species that are highly susceptible to DddA and others that are completely resistant to the toxin. This suggests that the determinants of cell fate following intoxication by DddA frequently vary between species. DNA repair pathways may provide one relevant source of diversity. Although our work did not implicate BER in susceptibility to DddA, we cannot rule out that the documented differences within this pathway could be responsible for at least part of the variability we observe (*van der Veen and Tang, 2015*). Our data show that Ung effectively removes the uracil resulting from DddA activity. However, the bulk uracil measurement methods we employed may not capture small changes in uracil levels distributed across bacterial chromosomes, or localized hotspots of uracil accumulation. Thus, it remains possible that levels of other glycosylases that act on uracil to initiate BER, such as mismatched uracil glycosylase (Mug), may influence cell fate following exposure to DddA (*Mokkapati et al., 2001*). Interestingly, *mug* orthologs are not universal among bacteria; the gene is present across Enterobacteriaceae, but it is not found in most Pseudomonads (*Brunder and Karch, 2000*; *Silby et al., 2011*). Another pathway that varies widely in bacteria and is implicated in uracil removal is mismatch repair (MMR). This pathway does not play a role in the removal of misincorporated uracils; however, those that derive from cytosine, and thus generate a mismatch, are potential substrates (*Krokan et al., 2014*). There is strong evidence linking MMR to uracil removal from U•G mismatches in eukaryotes, but the sparsity of comparable data in bacterial systems renders it challenging to estimate the extent to which MMR influences susceptibility to DddA (*López-Olmos et al., 2012*).

Other explanations for the variable effects of DddA on recipient cells may be unrelated to DNA repair entirely. Killing of target cells by interbacterial antagonistic systems can rely on the delivery of exceedingly small numbers of toxin molecules – in some cases as few as one (*Cascales et al., 2007*). We posit that the dependence on so few proteins could leave the fate of recipient cells subject to stochastic behavior. Recognition and turnover of DddA via cellular proteolytic machinery is one potential source of this stochasticity. In the majority of cells belonging to a resistant species, proteolytic machinery may fortuitously and effectively deplete DddA; however, in a small subset of these cells, this machinery might fail to act before DddA installs one or more mutations. In such a scenario, a species sensitive to the toxin would be expected to lack a pathway for DddA degradation altogether. This offers an explanation for our inability to detect mutations within surviving cells of bacteria sensitive to DddA; these may constitute a subpopulation that was not directly exposed to DddA. It is also possible that across a broader range of species, there exists a continuum of DddA

intoxication outcomes. Interbacterial competition studies between *B. cenocepacia* and a more expansive range of species will be informative in this regard.

Based on the inferred relationship between SsdA and DYW proteins, we were surprised to find that SsdA targets DNA, and not RNA (*Iyer et al., 2011*). Our data suggest that there may be a mosaic of substrate specificities even within relatively closely related clades of deaminases. Analyses performed by Aravind and colleagues provide support for the evolutionary origins of DYW proteins in bacteria (*Iyer et al., 2011*). Although there is substantial evidence implicating DYW proteins in RNA editing, little of this is direct and to our knowledge the activity of only one DYW protein has been reconstituted in vitro (*Hayes and Santibanez, 2017*). Therefore, it will be of interest to better understand whether DYW family members target RNA exclusively, or whether some might play roles in DNA editing. In this regard, our structure of the SsdA–SsdA$_I$ complex likely provides some insights. For example, it places the PG motif region, which differs markedly between BaDTF2 and DYW proteins, in position to engage substrate. Our structure also permits an initial comparison between bacterial deaminase toxins that target dsDNA (DddA) versus ssDNA (SsdA). Despite the predicted shared ancestry of DddA and SsdA, DddA displays significantly greater structural relatedness to eukaryotic deaminases targeting ssDNA (AID/APOBEC) than it does to SsdA. Taken together, this suggests that cytosine deaminases targeting ssDNA evolved at least twice from the ancestral deaminase fold (*Iyer et al., 2011*). Finally, the differing target specificities of DddA and SsdA suggest that both ds and ssDNA may be effective targets for toxic deaminases. Given that ssDNA represents only a fraction of total genetic material typically present in a bacterial cell, this finding seems counterintuitive. One explanation may be that because highly transcribed genes are more likely to exist in a single-stranded state, the toxicity of SsdA could result from the targeting of these specific regions. Alternatively, the lower level of dsDNA targeting by SsdA that we observe in vitro may represent the physiologically relevant activity of the protein during cellular intoxication.

While nutrient deprivation typically results in growth arrest, thymine starvation induced through genetic inactivation or chemical inhibition of thymine biosynthesis leads to cell death in organisms from *E. coli* to humans, in a process known as TLD (*Khodursky et al., 2015*; *Hanawalt, 2015*). There remains no consensus on the precise mechanism of TLD, but it is nevertheless possible to draw parallels between TLD and DddA intoxication. Both processes involve the incorporation of uracil into DNA and produce complex phenotypes including DNA replication arrest and chromosome degradation (*Khodursky et al., 2015*). Also, for both TLD and DddA intoxication, the removal of uracil incorporated into DNA does not rescue cells from intoxication (*Sangurdekar et al., 2010*). More generally, suppressor mutants resistant to killing by TLD appear to be difficult to obtain (*Fonville et al., 2011*; *Guzmán and Martín, 2015*). Our finding that Ung inactivation alleviates the phenotype of nucleoid degradation, yet Ung-deficient strains remain as sensitive to intoxication as the wild type, indicates that redundancy built into the mechanism by which DddA kill cells could similarly prevent suppressor mutant emergence. For TLD, this property has enabled the development of anticancer and antimicrobial drugs that induce TLD by acting as thymine analogs (*Thomson and Lamont, 2019*; *Tsesmetzis et al., 2018*); for DddA, difficulties associated with developing resistance to toxin activity may explain why toxins predicted to act as deaminases have become prevalent in the bacterial kingdom (*Iyer et al., 2011*).

Our studies of the mutagenic potential of deaminase toxins suggest these proteins may play an important role in the generation of diversity in microbial communities. Although researchers have overcome many hurdles to deciphering the intricate mechanistic details of interbacterial toxin delivery systems in vitro, it has remained difficult to extrapolate these results to the broader impact of the systems on bacterial ecology. In this regard, the indelible signature that deaminase toxins leave within the genomes of target cells offers a unique opportunity. Future studies aimed at mining genomic and metagenomic datasets for sequence signatures indicative of deaminase toxin activity have the potential to provide heretofore unobtainable insights into the complexities of bacterial interactions in nature.

## Materials and methods

### Bacterial strains and culture methods

Unless otherwise noted, bacterial strains used in this study were cultivated in Lysogeny Broth (LB) at 37°C with shaking or on LB medium solidified with agar (LBA, 1.5% w/v, except as noted). When necessary, antibiotics were supplied to the media in the following concentrations: carbenicillin (150 μg ml−1), gentamycin (15 μg ml−1), trimethoprim (50 μg ml−1), chloramphenicol (25 μg ml−1), irgasan (50 μg ml−1), kanamycin (50 μg ml−1), and streptomycin (50 μg ml−1). *E. coli* strains DH5α, XK1502, and BL21 were used for plasmid maintenance, toxicity, and mutagenesis assays, and protein expression, respectively. A detailed description of the remaining bacterial strains and plasmids used in this study is provided in the Key Resources Table.

### Molecular biology techniques and plasmid construction

All primers used in this study are listed in the Key Resources Table. The molecular biology reagents for DNA manipulation, Phusion high fidelity DNA polymerase, restriction enzymes, and Gibson Assembly Reagent were acquired from New England Biolabs (NEB). GoTaq Green Master Mix was obtained from Promega. Primers and gBlocks used in this study were acquired from Integrated DNA Technologies (IDT). Expression constructs for DddA$_{tox}$ (I35_RS34180) and DddA$_I$ were previously described (*Mok et al., 2020*). To generate a pETDuet-1-based expression construct for SsdA$_{tox}$ and SsdAI, the corresponding gene fragment and complete gene were amplified from *Pseudomonas syringae* and cloned into the MCS-1 (BamHI and NotI sites, introducing an N-terminal hexahistidine tag) and MCS-2 (NdeI and XhoI sites) using Gibson assembly, respectively. For deaminase toxicity assays, *ssdA*$_{tox}$ was amplified from *P. syringae* and the toxin domain of candidate BaDTF3 deaminase EL142_RS06975 of *Taylorella equigenitalus* was obtained by gBlock synthesis. Each were cloned individually into pSCRhaB2 (NdeI and XbaI sites) by Gibson assembly. The corresponding immunity genes *ssdA*$_I$ and EL142_RS06970 were also amplified and synthesized, respectively, and then cloned by Gibson assembly into pPSV39 (SacI and HindIII sites). A vector for markerless in-frame deletion of *ung* in *P. aeruginosa* PAO1 was generated by amplifying and combining 600 bp regions flanking the *ung* gene in the pEXG2 vector using PCR and Gibson assembly, generating pEXG2::Δ*ung*.

### Construction of genetically modified *P. aeruginosa*

A markerless in-frame deletion of *ung* in *P. aeruginosa* PAO1 was generated by allelic exchange with the suicide vector pEXG2::*ung*, employing SacB-based counter selection (*Rietsch et al., 2005*). The pEXG2::Δ*ung* vector was transformed into *E. coli* SM10 for conjugation into *P. aeruginosa*. Conjugation was performed by incubating a 1:1 mixture of *E. coli* SM10 (pEXG2::Δ*ung*) with *P. aeruginosa* for 6 hr on LBA. Selection for chromosomal integration of pEXG2::*ung* in *P. aeruginosa* was achieved by plating on LBA supplemented with irgasan and gentamycin. Resulting merodiploids were grown overnight, then plated on LBA supplemented with 5% (w/v) sucrose for SacB counter selection. Deletion of *ung* in resulting gentamycin susceptible colonies was confirmed by PCR.

### Construction of genetically modified *E. coli*

Deletions in *E. coli* were generated with Lamda-Red recombination (*Datsenko and Wanner, 2000*). Deletion cassettes for *nfo* and *xthA* were generated by amplifying the chloramphenicol resistance gene from pKD3 and kanamycin from pKD4, respectively, adding 50 base pairs of the region flanking the deletion target to the amplicon. Expression of the recombinase in *E. coli* carrying pKD46 was induced by sub-culturing an overnight culture of the strain in a 1:100 dilution in LB at 30°C with the addition of 20 mM arabinose. At OD$_{600}$ 0.6, the cells were recovered, washed repeatedly with sterile water then transformed by electroporation with the deletion cassettes. Successful deletion of the targeted genes was confirmed by antibiotic resistance and PCR. Both mutations were combined in a single strain by P1 phage transduction (*Thomason et al., 2007*). Lysates of *E. coli nfo*::cm were prepared by sub-culturing and overnight culture of the strain diluted 1:100 in LB until OD$_{600}$ 0.6, at which point different concentrations of P1 were added to the culture samples. After 1 hr of incubation, cultures that exhibited strong lysis were used for phage lysate preparation. Cell debris was removed by centrifugation and droplets of chloroform were used to remove any viable cells

remaining. Transduction was performed by combining an overnight culture of *E. coli xthA*::kan with phage lysate at different rations (1:1, 1:10, 1:100) in a final volume of 200 µL and incubating for 30 min at 37°C. Transduction was stopped with the addition of 100 µL 1 M Na-Citrate (pH 5.5) followed by an additional 30 min incubation at 37°C. Cells were then plated onto LB agar supplemented with kanamycin, choloramphenicol, and 100 mM Na-Citrate. Transductants were confirmed by PCR.

## Nucleoid and genomic DNA integrity assays

For fluorescence microscopy-based nucleoid integrity assays, overnight cultures of *E. coli* or *P. aeruginosa* strains were grown with the appropriate antibiotic. The cells were then subcultured by diluting 1:10 into fresh media, and incubated until OD$_{600}$ = 0.6, then *E. coli* cultures were supplemented with 0.2% rhamnose and *P. aeruginosa* cultures were supplemented with 1% arabinose followed by 1 additional hour of incubation. Cells were recovered and fixed in 4% formaldehyde for 15 min on ice, followed by a wash and resuspension in PBS. The resuspended cells were stained with DAPI (3 µg/ml) for 15 min then transferred to a 2% agarose pad in PBS for visualization. Fixed cells were imaged under a Nikon widefield microscope with a ×60 oil objective. For each condition, individual cells were segmented and quantified from three fields of view using SuperSegger software (*Stylianidou et al., 2016*). We obtained the average fluorescence intensity per pixel of each cell. A histogram of these averages yielded a bimodal distribution, from which we determined a cutoff value that separated degraded and intact nucleoids.

Cultures for gel electrophoresis-based gDNA integrity assessment were prepared in a similar way, but with the addition of sample collection after 0, 30, and 60 min after induction. A total of OD$_{600}$ = 0.3 in 1 ml of cells was pelleted and gDNA was extracted using DNeasy Blood and Tissue kit (Qiagen) using the Gram-negative bacterial protocol followed by quantification using Qubit. A total of 100 ng of gDNA was used for integrity visualization by 1% agarose gel electrophoresis, followed by with ethidium bromide staining imaging with an Azure C600.

## Fluorescence and phase-contrast microscopy of YPet-labeled DnaN

Prior to imaging, *E. coli* encoding YPet-labeled DnaN and carrying pSChRaB2::*dddA$_{tox}$* or empty vector and pPSV39::*dddI$_A$* were grown overnight with the appropriate antibiotics and 160 µM IPTG to induce immunity gene expression. Cells were then back diluted into M9 minimal medium (1X M9 salts, 2 mM MgSO4, 0.1 mM CaCl2, 0.2% glycerol, 10 µg/ml of thiamine hydrochloride, and 100 µg/ml each of arginine, histidine, leucine, threonine, and proline) with IPTG and grown in a bacterial tissue culture roller drum incubated at 30°C until reaching OD600 = 0.1. The cultures were then washed twice in the base growth medium (without IPTG) to lessen residual immunity expression. A volume of 2 µl was spotted onto a thin 2% (w/v) low-melt agarose (Invitrogen UltraPure LMP Agarose) pad composed of the base growth medium and 0.2% rhamnose for toxin induction. The sample was sealed on the pad under a glass cover slip using VaLP (a 1:1:1 Vaseline, lanolin, and paraffin mixture). Microscopy images were acquired using a Nikon Ti-E inverted microscope fitted with a ×60 oil objective (Nikon CFI Plan Apo Lambda 60X Oil), automated focusing (Nikon Perfect Focus System), a mercury arc lamp light source (Nikon IntensilightC-HGFIE), a sCMOS camera (Andor Neo 5.5), a custom-built environmental chamber, and image acquisition software (Nikon NIS Elements). The samples were imaged at 30°C at 5 min intervals over the duration of 6 hr. Cells were segmented in the time-lapse movies using SuperSegger (*Stylianidou et al., 2016*), a MATLAB-based image processing and analysis package, with focus tracking enabled (*Mangiameli et al., 2017*). Data points for the mean focus intensity plots (*Figure 2D*) were generated by averaging the scores of the single brightest focus of each segmented cell in each frame and then averaging that value over six fields of view, with the error bars representing the standard deviation of the final average. For the intoxicated strains, cell counts in the final frame fell within 300–600 for each field of view (FOV). For strains with the empty vector, final counts fell within 600–1500. Focus scores larger than 30 were excluded as outliers. After obtaining the data points, a centered moving average with nine points in the sample window was used to smooth out fluctuations.

## Bacterial competition and mutation frequency experiments

The fitness of *B. cenocepacia* strains in interbacterial interactions was evaluated in coculture growth assays. *B. cenocepacia* and competitor strains were grown overnight then concentrated to reach

OD$_{600}$ of 400 (*B. cenocepacia*) or 40 (competitors). The resulting cell suspensions were mixed in a 1:1 ratio for each *B. cenocepacia*-competitor pair, and 10, 10 µl samples of each mixture were spotted on a 0.2 µm nitrocellulose membrane placed on LBA (3% w/v) and then incubated 6 hr incubation at 37°C, or for 30, 90, and 150 min for time-course growth competition assays. After incubation, cells were recovered from the membrane surface and resuspended in 1 ml LB. The initial and final *B. cenocepacia* to competitor ratios were determined by diluting and plating fractions of the cultures on LBA with antibiotics selective for each organism. Mutation frequency was measured by plating post-incubation cultures onto LBA supplemented with rifampicin and an antibiotic to select for the non-*B. cenocepacia* competitor. Mutation frequency for the competitor species was then calculated by dividing the number of rifampicin-resistant colony-forming unit (cfu) by the total number of cfu recovered of that organism.

## *rpoB* and WGS of rifampicin-resistant colonies

Rifampicin-resistant colonies obtained from competition experiments were used for *rpoB* and WGS. For *rpoB* sequencing, the gene was amplified from resistant clones by colony PCR, and PCR amplicons were used as templates for Sanger sequencing. For WGS, cultures of isolates in LB were used for gDNA extraction using the Gram-negative bacterial protocol for the DNeasy Blood and Tissue kit (Qiagen). Extraction yield was quantified using a Qubit (Thermo Fisher Scientific). Libraries were constructed using the Nextera DNA Flex Library Prep Kit (Illumina) according to the manufacturer's protocol. Library concentration and quality was evaluated with a Qubit and 1% agarose gel electrophoresis. An Illumina iSeq (300 cycles paired-end program) was used for sequencing. Reads were mapped to reference genomes using the BWA software (*Li and Durbin, 2009*); reference genomes employed were NC_000913.3 for *E. coli* AB1157, NC_002655.2 for *E. coli* EHEC, CP000647.1 for *Klebsiella pneumoniae*, and NC_002516.2 for *P. aeruginosa* PAO1. Pileup data from alignments were generated with SAMtools, and variant calling was performed with VarScan (*Koboldt et al., 2012*; *Li, 2011*). For SNP calling, SNPs were considered valid if they had a frequency greater than 90%, with p-value<0.05, and if they were not present with parental strain.

## SNV analysis

SNV analysis was performed for both *E. coli* cultures intoxicated by heterologous expression of candidate deaminases (see *E. coli* toxicity assays) and for *P. aeruginosa* grown in coculture with *B. cenocepacia* (see Bacterial competition and mutation frequency experiments). For *E. coli*, 1 ml of pelleted cells subjected to intoxication was employed for genomic DNA extraction; for *P. aeruginosa*, cells were collected from 10, 10 µl competition mixtures prepared and grown as described above for 1 hr. In each case, gDNA extraction was performed with the DNeasy Blood and Tissue kit (Qiagen) using the Gram-negative bacterial protocol. Extraction yield was quantified using a Qubit (Thermo Fisher Scientific). Sequencing libraries were constructed using the Nextera DNA Flex Library Prep Kit (Illumina) as recommended by the manufacture, except the Enhanced PCR Mix (EPM) was substituted with KAPA HiFi HotStart Uracil+ ReadyMix (Kapa Biosystems) to enable the amplification of uracil encountered in the DNA template. Library concentration and quality was evaluated with a Qubit and 1% agarose gel electrophoresis. Sequencing was performed with an Illumina iSeq (300 cycles paired-end program), the BWA software was used to map reads against a reference genome (NC_000913.3) (*Li and Durbin, 2009*). For *P. aeruginosa* cocultures, the reads belonging to *P. aeruginosa* were separated from those derived from *B. cenocepacia* using the package BBmap (https://sourceforge.net/projects/bbmap/), and then BWA was used to map the reads against the *P. aeruginosa* reference genome (NC_002516.2) (*Li and Durbin, 2009*). For both *E. coli* and *P. aeruginosa* reads, Pileup data from alignments were generated with SAMtools, and variant calling was performed with VarScan (*Koboldt et al., 2012*; *Li, 2011*). We used a conservative threshold for SNV validation of variant frequency >0.005, coverage >50 reads per base, and p-value <0.05. Probability logos of the consensus region flanking modified bases were obtained by extracting the sequence in the position −3 to +3 flanking the SNV using custom Python scripts (https://github.com/marade/DeamData) (*de Moraes, 2020*), and the logo image was obtained by inputting the sequences in the WebLogo online tool (https://weblogo.berkeley.edu).

## Purification of Flag-ΔUNG-DsRed

Purification of Flag-tagged, catalytically inactive Ung fused to DsRed was performed as essentially as described in *Róna et al., 2016*. In brief, *E. coli* BL21(DE3) ung-151 carrying pET-20b::Flag-ΔUNG-DsRed was grown in LB broth to OD$_{600}$ 0.6 followed by the addition of 0.6 mM IPTG. The cultures were then incubated for 16 hr at 18°C. Cells were pelleted and resuspended in lysis buffer (50 mM TRIS·HCl, pH = 8.0, 300 mM NaCl, 0.5 mM ethylenediaminetetraacetic acid (EDTA), 10 mM β-mercaptoethanol, 1 mM phenylmethylsulfonyl fluoride, 5 mM benzamidine) followed by lysis by sonication. Supernatant was separated from debris by centrifugation at 20,000 × g for 30 min. Supernatant was then applied to a Ni-NTA column and washed with a series of buffers increasing in salt concentration (50 mM HEPES, pH = 7.5, 30 mM KCl, 5 mM β-mercaptoethanol; 50 mM HEPES, pH = 7.5, 300 mM KCl, 5 mM β-mercaptoethanol; 50 mM HEPES, pH = 7.5, 500 mM NaCl, 40 mM Imidazole, 5 mM β-mercaptoethanol). Flag-ΔUNG-DsRed was then eluted with elution buffer (50 mM HEPES, pH = 7.5, 30 mM KCl, 300 mM imidazole, 5 mM β-mercaptoethanol). Eluted sample was further purified with fast protein liquid chromatography (FPLC) and gel filtration on a Superdex200 column (GE Healthcare) in sizing buffer (30 mM Tris·HCl, pH = 7.4, 140 mM NaCl, 0.01% Tween-20, 1 mM EDTA, 15 mM β-mercaptoethanol, 5% (w/v) glycerol).

## Labeling of uracil in genomic DNA

Fluorescent labeling of uracil incorporated into genomic DNA was performed in *E. coli* expressing DddA or in *P. aeruginosa* cocultured with *B. cenocepacia*. For *E. coli,* overnight cultures of the strain carrying pSChRaB2::*dddA*$_{tox}$ or empty vector and pPSV39::*dddI*$_A$ were grown in overnight in LB with appropriate antibiotics and IPTG at 160 uM to induce immunity gene expression. Cultures were then diluted 1:100 into LB broth with antibiotics but without IPTG and grown until OD$_{600}$ = 0.6–0.8. Toxin expression was then induced with the addition of 0.2% rhamnose for 60 min, following which cells were collected by centrifugation. For *P. aeruginosa* in coculture with *B. cenocepacia*, overnight cultures were concentrated to OD$_{600}$ 40 and mixed in a 1:10 ratio. The mixture was then spotted on 0.2 μM nitrocellulose membrane on LBA (3% w/v) and incubated for 37°C for 1 hr. The cells were recovered and used for staining.

Uracil labeling with Flag-ΔUNG-DsRed was performed essentially as described in *Róna et al., 2016*. Cells collected from the experiments described above were resuspended in Carnoy's fixative reagent (ethanol 60%, methanol 30%, and chloroform 10%), then incubated 20 min at 4°C. Cells were rehydrated at room temperature by a series of 5 min incubations with buffer containing decreasing concentrations of ethanol (1:1 ethanol:PBS, 3:7 ethanol:PBS, then 100% PBS containing 0.05% Triton X-100 [PBST]). For permeabilization, cells were washed once with GTE buffer (50 mM glucose, 20 mM Tris, pH = 7.5, and 10 mM EDTA), then resuspended in GTE buffer containing 10 mg/ml lysozyme and incubated 5 min at room temperature. Lysozyme solution was removed by washing with PBST for 10 min followed by incubation with blocking buffer (5% BSA, in PBST) for 15 min. Genomic uracil residues were labeled by adding 5 μg/ml of purified Flag-ΔUNG-DsRed in blocking buffer and incubating for 1 hr at room temperature. Cells were again washed with PBST and then applied to a thin 2% agarose PBS slab on a microscopy slide for visualization. Quantification of uracil labeling was performed via Fiji (*Schindelin et al., 2012*) by calculating the mean fluorescence intensity of randomly selected ROIs (10 × 10 μm) in each FOV.

## Mass-spectrometry-based quantification of genomic uracil

Genomic uracil content was quantified for DNA extracted from cocultures of *B. cenocepacia* and *P. aeruginosa,* inoculated and grown as described above for bacterial competition assays, except cells were recovered after 1 hr of growth. Total genomic DNA extraction was performed using the DNeasy Blood and Tissue kit (Qiagen), and uracil quantification was performed as described previously using the excision method (*Galashevskaya et al., 2013*). Briefly, reactions containing 30 μg of gDNA and 1 U UNG in UNG buffer (20 mM Tris-HCl pH 7.5, 10 mM NaCl, 1 mM DTT, 1 mg/ml BSA) were incubated for 1 hr at 37°C for DNA deuracilation. Uracil (1,3-$^{15}$N$_2$) was then added to the reactions as an internal control at 2 nM, the reactions were extracted with 500 μl acetonitrile and centrifuged for 30 min at 14,000 rpm. The supernatant was then recovered, vacuum centrifuged until dried, followed by resuspension in 40 μl 10% 2 mM ammonium formate, 90% acetonitrile. Ten μl samples of the resuspensions were separated on a Waters Xbridge BEH amide column (2.5 μm, 130

Å, 2.1 × 150 mm) at 40°C using the following gradient at 0.300 ml/min: 0–3 min 95% B and 5% A, 3–8 min 95–50% B, and 5–50% A, 8–12 min 50% B and 50% A, 12–13 min 50–95% B and 50–5% A, 13–18 min 95% B and 5% A. Solvent A consisted of 95% water, 3% acetonitrile, 2% methanol, 0.2% Acetic acid (v/v/v/v) 10 mM ammonium acetate, and pH approximately 4.2 and solvent B consisted of 93% acetontrile, 5% water, 2% methanol, 0.2% acetic acid, and 10 mM ammonium acetate. The column was re-equilibrated between samples for 18 min at 95% B. Samples were then analyzed on an LC-QQQ system consisting of Shimadzu Nexera XR LC 20 AD pumps coupled to a Sciex 6500+ triple quadrupole mass spectrometer with Turbo V Ion Source operating in MRM mode through the Analyst 1.6.3 software. Curtain gas was at 9, source temperature was at 425°C, ionspray voltage was −4000V, GS1 was 70 and GS2 was 80. Uracil concentrations were quantified using Multiquant 3.0 software, and the internal control was used to account for variability during extraction and sample preparation.

## *E. coli* toxicity assays

*E. coli* strains carrying pSCRhaB2 expression constructs for deaminase toxins and pPSV39 expression constructs for the corresponding immunity determinants (*Figure 5*) were grown overnight in LB with the appropriate antibiotics and IPTG at 160 µM to induce immunity gene expression. Cultures were then diluted 1:100 into fresh medium without IPTG, incubated until OD600 = 0.6, then supplemented with 0.2% rhamnose for toxin induction. For time-course toxicity assays with DddA, aliquots of cultures were recovered at 0, 20, 40, 60, and 120 min after induction, plated onto LBA for cfu enumeration. For single time point assays with SsdA and EL142_RS06975, the cultures were plated on LBA at the time of toxin induction and after 60 min incubation, and proliferation was reported as final over initial cfu ratio.

## Purification of SsdA$_{tox}$ and SsdA$_{tox}$-SsdA$_I$ for biochemical assays and structure determination

For purification of his-tagged SsdA$_{tox}$ in complex with SsdA$_I$, an overnight cultures of *E. coli* BL21 (pETDuet-1::*ssdA$_{tox}$* + *ssdA$_I$*) was used to inoculate 2 l of LB broth in a 1:100 dilution. This culture was then grown to approximately OD$_{600}$ = 0.6, then induced with 0.5 mM IPTG and incubated for 16 hr at 18°C with shaking. Cell pellets were collected by centrifugation at 4000 g for 30 min, followed by resuspension in 50 ml of lysis buffer (50 mM Tris-HCl pH 8.0, 500 mM NaCl, 10 mM imidazole, 1 mg ml$^{-1}$ lysozyme, and protease inhibitor cocktail). Cell pellets were then lysed by sonication (five pulses, 10 s each) and supernatant was separated from debris by centrifugation at 25,000 g for 30 min. The his-tagged SddA$_{tox}$–SddI$_A$ complex was purified from supernatant by FPLC using a HisTrap HP affinity column (GE). Proteins were eluted with a final concentration of 300 mM imidazole. The eluted SddA$_{tox}$–SddI$_A$ complex was further subjected to gel filtration on a Superdex200 column (GE Healthcare) in sizing buffer (20 mM Tris-HCl pH 7.5, 200 mM NaCl). The fraction purity was evaluated by SDS-PAGE gel stained with Coomassie Brilliant Blue and the highest quality factions were stored at −80°C.

For SsdA$_{tox}$ separation from SsdA$_I$ to be used in biochemical assays, the elution underwent a denaturation and renaturation process. The elution was added to 50 ml 8 M urea denaturing buffer (50 mM Tris-HCl pH 7.5, 500 mM NaCl and 1 mM DTT) and incubated for 16 hr at 4°C. This suspension was then loaded on a gravity-flow column packed with 2 ml Ni-NTA agarose beads equilibrated with denaturing buffer. The column was washed with 50 ml 8 M urea denaturing buffer to remove unbound SsdA$_I$. On-column refolding of SsdA$_{tox}$ was achieved with sequential washes with 25 ml denaturing buffer with decreasing concentrations of urea (6 M, 4 M, 2 M, 1 M), and a last wash with wash buffer (50 mM Tris-HCl pH 7.5, 500 mM NaCl and 1 mM DTT) to remove remaining traces of urea. Refolded proteins bound to the column were then eluted with 5 ml elution buffer. The eluted samples were purified again by size-exclusion chromatography using an FPLC with gel filtration on a Superdex200 column (GE Healthcare) in sizing buffer (20 mM Tris-HCl pH 7.5, 200 mM NaCl, 1 mM DTT, 5% (w/v) glycerol). The fraction purity was evaluated by SDS–PAGE gel stained with Coomassie blue and the highest quality factions were stored at −80°C.

## Crystallization and structure determination

Crystals of the selenomethionine derivative of hexahistidine-tagged SsdA$_{tox}$ (a.a. 260–410)- SsdA$_I$ complex were obtained at 10 mg/ml in buffer containing 20 mM Tris pH 7.5, 200 mM NaCl, and 1.0 mM DTT. Crystallization was set up at room temperature by mixing the complex in this buffer 1:1 with crystallization buffer (20% [w/v] PEG 3350, 0.1 M Bis-Tris:HCl pH 7.5 and 200 mM MgCl$_2$). Long rod crystals were obtained in 2–5 days. Selenomethionine SsdA$_{tox}$-SsdA$_I$ crystals displayed the symmetry of space group I4$_1$22 (a = 93.07 Å, b = 93.07 Å, c = 383.49 Å, a = b = γ = 90°) (*Supplementary file 2*). Crystals were cryoprotected in a final concentration of 20% glycerol in crystallization buffer.

Diffraction datasets were collected at the BL502 beamline (ALS, Lawrence Berkeley National Laboratory). Data were indexed, integrated, and scaled using HKL-2000 *Otwinowski and Minor, 1997*. The calculated Matthews coefficient is Vm = 5.17, with 76% of solvent in the crystal and one molecule in an asymmetric unit. Coordinates and structure factors for SsdA/SsdA$_I$ complex have been deposited in the Protein Data Bank (PDB) under accession code 7JTU.

## DNA deamination assays

All DNA substrates were acquired from IDT, and contained a 6-FAM fluorophore for visualization. For assays shown in *Figure 5I and J*, substrates contained cytosine in each possible dinucleotide sequence context flanked by polyadenines. For the substrate preference assays shown in *Figure 5K*, substrates contained single cytosine residues, one substrate for each of the four possible upstream nucleotide contexts. The complete sequence for substrates is found in the Key Resources Table. To generate dsDNA substrates, a reverse complement oligo was annealed to the substrate containing the 6-FAM fluorophore. Reactions were performed in 10 µl of deamination buffer consisting of 20 mM Tris-HCl pH 7.4, 200 mM NaCl, 1 mM DTT, 1 µM substrate. The SsdA$_{tox}$ or controls APOBEC3A and DddA$_{tox}$ were added at the concentrations indicated in *Figure 5*. Reactions were incubated for 1 hr at 37°C, followed by the addition of 5 µl of UDG reaction solution (New England Biolabs, 0.02 U µl−1 UDG in 1X UDG buffer) and an additional 30 min incubation at 37°C. Cleavage of abasic sites generated by UDG-mediated cleavage of uracil residues in substrates was induced by addition of 100 mM NaOH and incubation at 95°C for 2 min. Reactions were analyzed by 15% acrylamide 8M urea gel electrophoresis in TBE buffer and the 6-FAM fluorophore signal was detected by fluorescence imaging with an Azure C600.

## Poisoned primer extension assay for RNA deamination

The RNA substrates and the DNA oligonucleotide containing a 5′ 6-FAM fluorophore for visualization (Key Resources Table) were acquired from IDT. The RNA substrates were designed to allow the 3′ end of the DNA oligonucleotide to anneal immediately before the cytidine target for deamination. Deamination reactions were performed in deamination buffer (20 mM Tris-HCl pH 7.4, 200 mM NaCl, 1 mM DTT), with the addition of 1 µM substrate and SsdA$_{tox}$, or the negative control protein DddA$_{tox}$ according to the concentrations shown in *Figure 5—figure supplement 1*. Substrate combinations and concentrations were added as indicated in *Figure 5—figure supplement 1*, and reactions were incubated for 1 hr at 37°C. cDNA synthesis was then performed in a 10 µl reaction (2.5 U µl−1 MultiScribe Reverse Transcriptase (Thermo Fisher), 1 µl deamination reaction, 1.5 µM oligonucleotide, 100 µM ddATP, 100 µM dCTP, 100 µM dTTP, and 100 µM dGTP) incubated at 37°C for 1 hr. Samples were analyzed by denaturing 15% acrylamide eight urea gel electrophoresis in TBE buffer. The synthesized cDNA fragments were detected by fluorescence imaging with an Azure Biosystems C600.

## Statistics

Student's t-test analyses were performed using GraphPad Prism version 8.0. To test for enrichment for C•G-to-T•A mutations in the context preferred by DddA in intoxicated populations (experimental condition) when compared to the pattern of mutations found to arise spontaneously in *E. coli* under neutral selection (control condition), a fisher's exact test, and negative binomial regression were employed. The fisher's exact test compared presence-absence of any C•G-to-T•A mutations in the context preferred by DddA outside of *rpoB*, as a binary variable for each clone in the experimental and control conditions. The negative binomial regression compared counts of C•G-to-T•A mutations

in the context preferred by DddA outside of *rpoB* between the experimental and control conditions. The [log] overall mutation rate per condition was included as an offset in the negative binomial regression to make the conditions as comparable as possible. These analyses were run using R version 3.6.2.

### Sequencing data
Sequencing data associated with this study have been deposited at the NCBI Trace and Short-Read Archive (SRA) under BioProject accession ID PRJNA659516.

## Acknowledgements

We thank Beata Vértessy and Reuben Harris for providing reagents, Simon Dove for critical review of the manuscript, and Ajai Dandekar and members of the Mougous laboratory for insightful discussions. DNA sequencing support was provided by the University of Washington Cystic Fibrosis Research Translation Center and Research Development Program Genomic Sequencing Core, which is funded by the National Institutes of Health (NIH, DK089507) and the Cystic Fibrosis Foundation (CFF, SINGH19R0). The work was supported by NIH grants GM128191 (to PAW) and AI080609 (to JDM). MHdM was supported by CFF Fellowship DEMORA18F0 and JDM holds an Investigator in the Pathogenesis of Infectious Disease Award from the Burroughs Wellcome Fund and is an Investigator of the Howard Hughes Medical Institute.

## Additional information

### Funding

| Funder | Grant reference number | Author |
| --- | --- | --- |
| National Institutes of Health | GM128191 | Paul A Wiggins |
| National Institutes of Health | AI080609 | Joseph D Mougous |
| Howard Hughes Medical Institute | | Joseph D Mougous |
| Cystic Fibrosis Foundation | DEMORA18F0 | Marcos H de Moraes |
| Burroughs Wellcome Fund | | Joseph D Mougous |
| National Institutes of Health | DK089507 | Marcos H de Moraes |
| Cystic Fibrosis Foundation | SINGH19RO | Marcos H de Moraes |

The funders had no role in study design, data collection and interpretation, or the decision to submit the work for publication.

### Author contributions

Marcos H de Moraes, Conceptualization, Data curation, Software, Formal analysis, Supervision, Validation, Investigation, Visualization, Methodology, Writing - original draft, Project administration, Writing - review and editing; FoSheng Hsu, Formal analysis, Investigation, Visualization, Methodology; Dean Huang, Software, Formal analysis, Investigation, Visualization, Methodology; Dustin E Bosch, Data curation, Formal analysis, Investigation; Jun Zeng, Jacob P Frick, Investigation; Matthew C Radey, Software, Investigation; Noah Simon, Validation; Hannah E Ledvina, Investigation, Methodology; Paul A Wiggins, Supervision, Methodology; S Brook Peterson, Conceptualization, Formal analysis, Supervision, Methodology, Writing - original draft, Project administration, Writing - review and editing; Joseph D Mougous, Conceptualization, Supervision, Funding acquisition, Visualization, Methodology, Writing - original draft, Project administration, Writing - review and editing

### Author ORCIDs

Dean Huang http://orcid.org/0000-0002-1235-5699
S Brook Peterson https://orcid.org/0000-0003-2648-0965
Joseph D Mougous https://orcid.org/0000-0002-5417-4861

Decision letter and Author response
Decision letter https://doi.org/10.7554/eLife.62967.sa1
Author response https://doi.org/10.7554/eLife.62967.sa2

# Additional files

## Supplementary files

• Supplementary file 1. Summary of single-nucleotide variants (SNPs) identified within target cells after co-culture with *B. cenocepacia*.

• Supplementary file 2. X-ray data collection and refinement statistics.

• Transparent reporting form

## Data availability

Diffraction data have been deposited in PDB under the accession code 7JTU. Sequencing data have been deposited at the NCBI Trace and Short-Read Archive (SRA) under BioProject accession ID PRJNA659516.

The following dataset was generated:

| Author(s) | Year | Dataset title | Dataset URL | Database and Identifier |
| --- | --- | --- | --- | --- |
| de Moraes MH, Radey MC | 2020 | An interbacterial toxin directly mutagenizes surviving target populations | https://www.ncbi.nlm.nih.gov/bioproject/?term=prjna659516 | NCBI BioProject, PRJNA659516 |

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

# Appendix 1

**Appendix 1—key resources table**

| Reagent type (species) or resource | Designation | Source or reference | Identifiers | Additional information |
|---|---|---|---|---|
| Strain, strain background (*Escherichia coli*) | DH5α | Thermo Fisher Scientific Cat#18258012 | F−φ 80*lacZ*ΔM15 Δ (*lacZYA-argF*) U169 *recA*1 *endA*1 *hsdR*17 (rK−, mK+) *phoAsupE44* λ−*thi-1 gyrA96relA1* | Used for cloning |
| Strain, strain background (*Escherichia coli*) | BL21 (DE3) | EMD Millipore Cat#69450 | F−*ompT hsdSB* (rB−, mB−)*gal dcm* (DE3) | Used for protein expression |
| Strain, strain background (*Escherichia coli*) | BL21 ung-151 | PMID:15096615 | BDSC:31777; FLYB: FBtp0001612; RRID:BDSC_31777 | Used for protein expression |
| Strain, strain background (*Escherichia coli*) | XK1502 | PMID:16923902 | F− *ompT hsdSB* (rB−, mB−) *gal dcm ung*-151 | Used for protein expression |
| Strain, strain background (*Escherichia coli*) | XK1502 Δung | PMID:32641830 | F− Δ*lac*U169 *nalA* Δ*ung* | Used for protein expression |
| Strain, strain background (*Escherichia coli*) | AB1157 | PMID:17248104 | F- thr-1 leuB6 (Am) glnX44 (AS) hisG4(Oc) rfbC1 rpsL31 (strR) argE3(Oc) | |
| Strain, strain background (*Escherichia coli*) | AB1157 γ*pet-dnaN* | PMID:28114307 | γ*pet-dnaN*, KanR | |
| Strain, strain background (*Escherichia coli*) | CJ236 | PMID:24723723 | F⁺*ung-1 relA1 dut-1 spoT1 thiE1* | |
| Strain, strain background (*Burkholderia cenocepacia*) | H111 | PMID:10713433 | Wild-type | |
| Strain, strain background (*Burkholderia cenocepacia*) | H111 ΔI35_RS01770 | PMID:32641830 | ΔI35_RS01770 | |
| Strain, strain background (*Burkholderia cenocepacia*) | H111 *dddA*$^{E1347A}$ | PMID:32641830 | *dddA*$^{E1347A}$ | |

*Continued on next page*

*Appendix 1—key resources table continued*

| Reagent type (species) or resource | Designation | Source or reference | Identifiers | Additional information |
|---|---|---|---|---|
| Strain, strain background (*Pseudomonas aeruginosa*) | PAO1 | PMID:10984043 | Wild-type | |
| Strain, strain background (*Klebsiella pneumoniae*) | MGH 78578 | PMID:11677609 | Wild-type | |
| Strain, strain background (*Escherichia coli*) | O157:H7 EDL933 | PMID:11206551 | | |
| Strain, strain background (*Acinetobacter baumannii*) | ATCC 17978 | PMID:18931120 | Wild-type | |
| Strain, strain background (*Burkholderia thailandensis*) | E264 | PMID:16336651 | Wild-type | |
| Strain, strain background (*Burkholderia thailandensis*) | F1 | PMID:23618999 | Wild-type | |
| Strain, strain background (*Burkholderia cenocepacia*) | K56-2 | PMID:29208119 | Wild-type | |
| Strain, strain background (*Salmonella enterica*) | Sv. Typhimurium 14028 s | PMID:19897643 | Wild-type | |
| Recombinant DNA reagent | pPSV39-CV (plasmid) | PMID:23954347 | | For inducible expression of proteins in *E. coli* |
| Recombinant DNA reagent | pScrhaB2-V (plasmid) | PMID:15925406 | | For inducible expression of proteins in *E. coli* |
| Recombinant DNA reagent | pEXG2 (plasmid) | PMID:15911752 | | For generation of markless *P. aeruginosa* mutants |
| Recombinant DNA reagent | pScrhaB2-V::*ssdA* (plasmid) | This study | | To express *ssdA* |
| Recombinant DNA reagent | pPSV39-CV::*ssdA$_I$* (plasmid) | This study | | To express *ssdA$_I$* |
| Recombinant DNA reagent | pScrhaB2-V:: *TequE* (plasmid) | This study | | To express BadTF3 |
| Recombinant DNA reagent | pPSV39-CV:: *TequE* (plasmid) | This study | | To express BadTF3-Imm |
| Recombinant DNA reagent | pETDuet-1 mcs1:: *ssdA-his$_6$* mcs1:: *ssdA$_I$* (plasmid) | This study | | To express *ssdA-ssdA$_I$* |
| Recombinant DNA reagent | pexG2_Δ*ung* (plasmid) | This study | | To delete ung in *Pseudomonas aeruginosa* |

*Continued on next page*

*Appendix 1—key resources table continued*

| Reagent type (species) or resource | Designation | Source or reference | Identifiers | Additional information |
|---|---|---|---|---|
| Sequence-based reagent | ungDel-1 | This study | | 5'-CAAGCTTCTGCAGGTCGACT CTAGAGGTATGGAGTTGTCCTTCGG |
| Sequence-based reagent | ungDel-2 | This study | | 5-AGAGGTCCGGATC GGTCATGGAACCCCC |
| Sequence-based reagent | ungDel-3 | This study | | 5'-CATGACCGATCCGGA CCTCTGAAGGCCGC |
| Sequence-based reagent | ungDel-4 | This study | | GGAAATTAATTAAGGTACCGAA TTCCCGCGCCGGTGGACTGGC |
| Sequence-based reagent | PAO1-ung-F | This study | | 5'-CCGGGGAGTACTTCTCGTTC |
| Sequence-based reagent | PAO1-ung-R | This study | | 5'-GGCGTTCCAGTACCTGCTC |
| Sequence-based reagent | GA_duet_PsyrE1-F | This study | | 5'-ACCATCATCACCACAGCCAGGAT CCGAAGGTCTCAAATATTGCG |
| Sequence-based reagent | GA_duet_PsyrE1-R | This study | | 5'-CTTAAGCATTATGCGGCCGCT CATTCCGACCTCATAATTG |
| Sequence-based reagent | GA_duet_PsyrI1-F | This study | | 5'-TATAAGAAGGAGA TATACATATGAATAA CAAAAGTAAAGTATTGATTGAAAAGC |
| Sequence-based reagent | GA_duet_PsyrI1-R | This study | | 5'-GCCGGCCGATATCCAATTGAGAT CTTCACACAACTTGCGGCAC |
| Sequence-based reagent | GA_pRhB_PsyrE1-F | This study | | 5'-TGAAATTCAGCAGGATCACATA TGAAGGTCTCAAATATTGCG |
| Sequence-based reagent | GA_pRhB_PsyrE1-R | This study | | 5'-TCATTTCAATATCTGTATATCTAGA TTCCGACCTCATAATTGTTTC |
| Sequence-based reagent | GA_p39_PsyrI1-F | This study | | 5'-ACAATTTCAGAATT CGAGCTCACGGGAGGAAA GATGAATAACAAAAGT AAAGTATTGATTGAAAAGC |
| Sequence-based reagent | GA_p39_PsyrI1-R | This study | | 5'-TCATTTCAATATCTGTATATCTA GATCACACAACTTGCGGCAC |
| Sequence-based reagent | rpoB-F | This study | | 5'-GGAAAACCAGTTCCGCGTTG |
| Sequence-based reagent | rpoB-R | This study | | 5'-TCCAAGTTGGAGTTCGCCTG |
| Sequence-based reagent | nfo_del-F | This study | | 5'-CATTACCGTTTTCCTCC AGCGGGTTTAACAGGAGTCCT CGCATGAAATACGTGT AGGCTGGAGCTGCTTC |
| Sequence-based reagent | nfo_del-R | This study | | 5'-CCGTAAAATTGCAAG GATCTCCTTTTCCCGGTTATT CATCTTCAGGCTACCATAT GAATATCCTCCTTAG |
| Sequence-based reagent | nfo_dt-F | This study | | 5'-GCTGATGGCACTGGTACTGT |
| Sequence-based reagent | nfo_dt-R | This study | | 5'-CCTTTAATCCGGCCTTTGCG |
| Sequence-based reagent | xthA_del-F | This study | | 5'-TACCATCCACGCACTCT TTATCTGAATAAATGGCAGCGA CTATGAAATTTGTGTA GGCTGGAGCTGCTTC |

*Continued on next page*

*Appendix 1—key resources table continued*

| Reagent type (species) or resource | Designation | Source or reference | Identifiers | Additional information |
|---|---|---|---|---|
| Sequence-based reagent | xthA_del-R | This study | | 5'-TTAATTCTCCTGACCCAG TTTGAGCCAGGAGAGCTGC TAAATTAGCGGCGCA TATGAATATCCTCCTTAG |
| Sequence-based reagent | xthA_dt-F | This study | | 5'-TACGTTTGCGATGTGGGTGA |
| Sequence-based reagent | xthA_dt-R | This study | | 5'-ATAACAAAGGACGGCAGGCA |
| Sequence-based reagent | EL142_RS06975 [tox]-gBlock | This study | | 5'-TGAAATTCAGCAGGATCACA TATGCTTTTAGGTGGAC TTAACAACTACCAATACGCCCC AAATCCAGTCGAATGGGT CGATCCTTTGGGTTGGAAATTCT CCAATGGCAAGCGTCG TCCGCCCCACAAGGCAACGGTT ACCGTCACAGACAAGAAC GGAGTGGTCAAACACAAATCCAA TTTGGTGTCAGGAAATATG ACAGAAGCCGAAAAAAAACTGGG TTTCCCGAACAACTCTTT GGCAACACATACCGAGAATCGTG CAACGCGCTTAATTGACC TGAATCAAGGTGATACCATGTTAA TTGAAGGACAGTATCGCC CGTGCCCACGCTGTAAGGGTGCG ATGCGTGTTAAGGCAGAG GAATCTGGGGCTAAGGTTAT CTACACCTGGCCCGAAGACGG TGACTTGAAGAAGCGCGAGT GGGAAGGAACCCCCTGT GATAAAAAGTCTAGAT ATACAGATATTGAAATGA |
| Sequence-based reagent | EL142_RS06970 -gBlock | This study | | 5'-GCCCCAAGGGGTTATGCTA AAGCTTTCTTAATTATCGAA CAGGAATTTCAGATGATTCTCG ACTTCAAGAATGTCAATCT TTTCAATCAAGCGAAGACGAA CGTAGTTATTAGGCTCAATT AAAAGATCGACACAATTTGGTT CGGACAGTAAATAGCCGGA CAAGCCATAGGGGTTAAAATCA AGGAACAGATTAGACTCCA GATAGATGTCTCCCTCGGGGAT ACCACCGCTGCGTTCCGTT GTTGAACGTGTAATGATGAAAT ACTGCTCAGGCACTTCATA GCTATCAGATGCAAACGAAAGG ATGGCGCAATAATCCTCAA TCGCAAACTTCACTTCTTGAA TCACGATGTTACTCAGCATC TTTCCTCCCGTGAGC TCGAATTCTGAAATTGT |
| Sequence-based reagent | RNA-GGCCGG | This study | | 5'-GAGGCCGGAAGUGGAU GUGGAUAAGAUGGAG |
| Sequence-based reagent | RNA-GGCUGG | This study | | 5'-GAGGCUGGAAGUGGAU GUGGAUAAGAUGGAG |
| Sequence-based reagent | PPE-Oligonucleotide | This study | | 5'FAM-CTCCATCTTATCCACATCCACT |
| Sequence-based reagent | DNA-ATGCGCCA | This study | | 5'FAM-AAAAAAAAAAAAA ATGCGCCAAAAAAAAAAAAAAAA |

*Continued on next page*

*Appendix 1—key resources table continued*

| Reagent type (species) or resource | Designation | Source or reference | Identifiers | Additional information |
|---|---|---|---|---|
| Sequence-based reagent | revDNA-ATGCGCCA | This study | | 5'-TTTTTTTTTTTTTTTTGGCGCATTTTTTTTTTTTTTTT |
| Sequence-based reagent | DNA-GCG | This study | | 5'-FAM-AAAAAAAAAAAAAAAAGCGAAAAAAAAAAAAAAAAA |
| Sequence-based reagent | DNA-CCG | This study | | 5'-FAM-AAAAAAAAAAAAAAAACCGAAAAAAAAAAAAAAAAA |
| Sequence-based reagent | DNA-TCG | This study | | 5'-FAM-AAAAAAAAAAAAAAAATCGAAAAAAAAAAAAAAAAA |
| Sequence-based reagent | DNA-ACG | This study | | 5'-FAM-AAAAAAAAAAAAAAAAACGAAAAAAAAAAAAAAAAA |
| Sequence-based reagent | revDNA-GCG | This study | | 5'-TTTTTTTTTTTTTTTTTCGCTTTTTTTTTTTTTTTT |
| Sequence-based reagent | revDNA-CCG | This study | | 5'-TTTTTTTTTTTTTTTTTCGGTTTTTTTTTTTTTTTT |
| Sequence-based reagent | revDNA-TCG | This study | | 5'-TTTTTTTTTTTTTTTTTCGATTTTTTTTTTTTTTTT |
| Sequence-based reagent | revDNA-ACG | This study | | 5'-TTTTTTTTTTTTTTTTTCGTTTTTTTTTTTTTTTT |
| Gene (*Pseudomonas syringae*) | *ssdA* | GeneBank | | PSYRH_RS14050 |
| Gene (*Pseudomonas syringae*) | *ssdAI* | GeneBank | | PSYRH_RS1404 |
| Gene (*Taylorella equigenitalis*) | BadTF3 | GeneBank | | EL142_RS06975 |
| Gene (*Taylorella equigenitalis*) | BadTF3-Immunity | GeneBank | | EL142_RS06970 |

