## [Decision Letter]

**Acceptance summary:**

This is an imaginative and broad study of effects of a newly discovered deaminase DddA in *Burholderia cenocepacia* on a variety of other bacteria, the role of its orthologs in other species, and a compelling survey of the previously unappreciated gene family. DddA deaminates cytosine to generate uracil, which is highly toxic to some bacteria, but other are resistant. This manuscript focuses on the identification and characterization of both the toxic and mutagenic capacity of DddA in susceptible and resistant cell populations, respectively. The authors show an increase in antibiotic resistance consistent with high mutagenesis rates. Both resistant and susceptible (those bacteria that survive competition) bacterial populations accumulate mutations. The authors go on to characterize additional classes of DddA like toxins and identify newly named SsdA1 from P. syringae as a DNA targeting deaminase. This study includes the structure of SsdA1 that indeed confirms its evolutionary relationship within this deaminase family, whose evolutionary origins are fascinating to ponder.

**Decision letter after peer review:**

Thank you for submitting your article "An interbacterial DNA deaminase toxin directly mutagenizes surviving target populations" for consideration by *eLife*. Your article has been reviewed by three peer reviewers, including Vaughn S Cooper as the Reviewing Editor and Reviewer #1, and the evaluation has been overseen by a Reviewing Editor and George Perry as the Senior Editor. The following individual involved in review of your submission has agreed to reveal their identity: Neal M Alto (Reviewer #3).

The reviewers have discussed the reviews with one another and the Reviewing Editor has drafted this decision to help you prepare a revised submission.

Summary:

This is an imaginative study of effects of a newly discovered deaminase DddA in *Burholderia cenocepacia* on a variety of other bacteria, the role of its orthologs in other species, and a compelling survey of the previously unappreciated gene family. As reported previously by this group, DddA exhibits deaminase activity and deaminates cytosine to generate uracil. This activity is highly toxic to some bacteria, but other are resistant. This manuscript focuses on the identification and characterization of both the toxic and mutagenic capacity of DddA in susceptible and resistant cell populations, respectively. Initially, this study shows genome instability catalyzed by DddA results from removal of Uracil by Uracil DNA glycosylase (Ung). However, this activity is not sufficient for bacterial killing by DddA. Despite a logical series of experiments, the mechanism of *E. coli* cell death by DddA remains unclear. However, the authors nicely rule some possible explanations. In an interesting twist, the authors show that T6SS delivery of DddA does not kill *E. coli*. Instead, they identify an increase in antibiotic resistance consistent with high mutagenesis rates. Both resistant and susceptible (those bacteria that survive competition) bacterial populations accumulate mutations. The authors go on to characterize additional classes of DddA like toxins and identify newly named SsdA1 from P. syringae as a DNA targeting deaminase. This study includes the structure of SsdA1 that indeed confirms its evolutionary relationship this deaminase family. Overall, the studies presented in the manuscript are high quality and rigorous. The article has great promise. However, the arguments that this system has evolved to increase genetic diversity are speculative and poorly founded, and generally weaken an otherwise strong paper. Several specific comments about this follow.

Essential revisions:

1) The research is largely conducted in trans in *E. coli*, and the native effects of this enzyme are largely unclear. The method of expressing dddA in *E. coli*, the extent to which results are a product of mismatch, and the justification to study in this organism rather than another, e.g. by overexpressing in Burkholderia, is unclear.

2) Can the authors estimate how different DddA levels are after heterologous expression and T6SS delivery? I realise the authors are aware of the difference, but an estimate on how different these levels are would be useful.

3) Why didn't you sequence WT genomes exposed for less time or concentration of DddA? Effect on ung background is very interesting as proof of principle but ung genotypes are absent in nature.

4) Please consider your findings in light of the work on the WT mutational spectrum of B. cenocepacia, which found elevated GC->TA transitions in late replicated regions: https://www.genetics.org/content/200/3/935

5) Subsection “Bacteria resistant to DddA-mediated intoxication accumulate DddA-catalyzed mutations”. All of the inhibitory activity due to DddA – this is an overstatement, you haven't possibly examined the range of conditions / environments by which T6SS could inhibit competitors.

6) Discussion. The manuscript focuses exclusively on toxic editing of the nucleic acids of competitors, not the origins of genetic diversity. This point is speculative and not well founded by the manuscript. If you wish to make this point, please consider the native effects of overexpressing these enzymes in B. cenocepacia, or wherever the ortholog is found.

7) Discussion. Again, see the article in which we actually measured mutation spectra in Burkholderia – a better reference than some cited. And further, studying mutation spectra in monocultures allows for a measure of the endogenous effects of traits like this.

8) Discussion. About multiple mutations arising concurrently. We see no evidence of this, again see Dillon et al., 2016.

9) Discussion. The speculation about the evolution of mutators related to this phenotype is unfounded and based on a fuzzy group selection argument. Bacterial populations are rarely starved for genetic variation, and when they are, defects in DNA repair might hitchhike with beneficial alleles. There is a great deal of theory and experimentation about the conditions for the evolution of mutators – see e.g. Raynes and Sniegowski, (2014) for a good review.

10) Subsection “Bacterial strains and culture methods”. I do not believe any evidence is presented that this mechanism can generate diversity other than by killing competitors.

Revisions expected in follow-up work:

The selection of organisms for competition experiments raises some questions. Why two *E. coli* strains? Why not include other Bcc organisms? In general, a broader taxonomic sampling would have been interesting and could have shed some light on ecological roles. It is e.g. tempting to assume that *B. cenocepacia* (a plant-associated organism turned opportunistic pathogen) experiences more competition in its natural habitat (rhizosphere) than in the lungs of CF patients. Or alternatively, its ability to kill other opportunistic pathogens may have allowed it to invade these lungs in the first place. I am not saying these questions are easy to answer, but a better global picture of DddA's spectrum of activity could be very informative.

One criticism is that the study tackles several different objectives, each of which is only partially explored. For example, determining why some cells are resistant and others are sensitive to DddA remains and open question. What role the mutagenesis activity of DddA and its homologs may play in non-laboratory situations is currently unknown (although difficult to examine). Finally, the virulence role of SsdA1 has not yet been elucidated.

[Editors' note: further revisions were suggested prior to acceptance, as described below.]

Thank you for submitting your article "An interbacterial DNA deaminase toxin directly mutagenizes surviving target populations" for consideration by *eLife*. Your article has been reviewed by three peer reviewers, including Vaughn S Cooper as the Reviewing Editor and Reviewer #1, and the evaluation has been overseen by George Perry as the Senior Editor. The following individual involved in review of your submission has agreed to reveal their identity: Neal M Alto (Reviewer #3).

The reviewers have discussed the reviews with one another and the Reviewing Editor has drafted this decision to help you prepare a revised submission.

Summary:

This is an imaginative study of effects of a newly discovered deaminase DddA in *Burkholderia cenocepacia* on a variety of other bacteria, the role of its orthologs in other species, and a compelling survey of the previously unappreciated gene family. As reported previously by this group, DddA exhibits deaminase activity and deaminates cytosine to generate uracil. This activity is highly toxic to some bacteria, but other are resistant. This manuscript focuses on the identification and characterization of both the toxic and mutagenic capacity of DddA in susceptible and resistant cell populations, respectively. Initially, this study shows genome instability catalyzed by DddA results from removal of Uracil by Uracil DNA glycosylase (Ung). However, this activity is not sufficient for bacterial killing by DddA. Despite a logical series of experiments, the mechanism of *E. coli* cell death by DddA remains unclear, although the authors nicely rule some possible explanations. In an interesting twist, the authors show that T6SS delivery of DddA does not kill *E. coli*. Instead, they identify an increase in antibiotic resistance consistent with high mutagenesis rates. Both resistant and susceptible (those bacteria that survive competition) bacterial populations accumulate mutations. The authors go on to characterize additional classes of DddA like toxins and identify newly named SsdA1 from P. syringae as a DNA targeting deaminase. This study includes the structure of SsdA1 that indeed confirms its evolutionary relationship this deaminase family. Overall, the studies presented in the manuscript are high quality and rigorous.

We appreciate the detailed and thoughtful responses to the prior round of reviews and find this revision significantly improved. This system is remarkable and its evolutionary origins are fascinating to ponder especially in light of its representation of a novel protein family. Yet there are a few remaining issues that need to be addressed before acceptance, as outlined below.

Essential revisions:

1) It appears that DddA can increase genetic variation by roughly a log in neighboring cells. and some of these mutations could in theory be beneficial to the neighbors in some circumstances. However, it's unclear (1) how this trait would be selected and maintained in one lineage for the express benefit of unrelated others and (2) how frequently the process of mutagenizing neighboring populations would allow them to adapt beyond their unaffected capacity.

2) On point 2: in theory, this effect would be strongest in small, clonal, structured populations as you suggest, and I agree that the unique effect of DddA might be its ability to produce multiple mutations per generation. However, I couldn't find clear evidence of multiple simultaneous mutations per cell in this study or the preceding publication. (This would likely require WGS of effectively single cells, which is tricky in this system.) Rather, results of cumulative accumulation of mutations over multiple generations of growth seem to be reported. Further, the measurement of mutation rates (e.g Figure 4) was also inferred from cumulative effects of population growth, in which jackpot effects of early-mutated cells could be amplified and lead to overestimates in mutation rates. This problem and appropriate measures are described here: https://aac.asm.org/content/52/4/1209. Please revisit your estimates of mutation rates in terms of mutational events, which require an assessment of Poisson variation among many replicates.

3) On point 1, this argument is featured in the Abstract, which indicates that this remains a major claim of the paper, but it's unfounded. The study presents no evidence that mutations acquired by affected cells are adaptive and increase fitness, and the conditions that might allow for evolution of the proposed altruistic effects are vanishingly rare in theory and absent in the literature. I recommend removal from the Abstract and further caution in the corresponding sentences in the Discussion.

---

## [Author Response]

Essential revisions:1) The research is largely conducted in trans in *E. coli*, and the native effects of this enzyme are largely unclear. The method of expressing dddA in *E. coli*, the extent to which results are a product of mismatch, and the justification to study in this organism rather than another, e.g. by overexpressing in Burkholderia, is unclear.

The reviewers are correct in pointing out that a number of our mechanistic studies of DddA activity were conducted using heterologous expression in *E. coli*. However, we respectfully wish to point out that our original submission included extensive data supporting our conclusion that its mechanism of toxicity during over-expression is similar when it is delivered to a susceptible host (*P. aeruginosa*) via the T6SS, as would naturally occur. Indeed, Figure 4 was devoted to this point. These data included: (i) uracil accumulation on the genome of *P. aeruginosa* during interbacterial competition with *B. cenocepacia* containing DddA (Figure 4A and Figure 4—figure supplement 1), (ii) mutation of *P. aeruginosa* ∆*ung* during interbacterial competition with *B. cenocepacia* containing DddA (Figure 4B-D), and *iii*) the demonstration that wild-type and ∆*ungP. aeruginosa* are equally susceptible to intoxication by DddA (Figure 4E).

Nevertheless, in our revised manuscript, we have added to these data, demonstrating that expression of DddA leads to nucleoid degradation in *P. aeruginosa* in a similar manner to its effect in *E. coli* (Figure 3—figure supplement 3). We have also reorganized the manuscript to feature these results more prominently (see Figure 3).

2) Can the authors estimate how different DddA levels are after heterologous expression and T6SS delivery? I realise the authors are aware of the difference, but an estimate on how different these levels are would be useful.

The precise levels of T6SS effectors delivered to target cells has been difficult to quantify, as these proteins are recalcitrant to fusions, present at low levels, and it is technically challenging to separate target and donor cell populations on a rapid enough time-scale to make meaningful measurements. It is generally believed that very few molecules of any individual effector are delivered to a recipient cell via the T6SS. Single delivery (sheath contraction) events can be monitored by time-lapse microscopy and after just one or two of these events a target cell can be observed succumbing to intoxication (e.g. Basler et al., 2013, Gerc et al., 2015). If our current structural model of T6SS is correct, each assembled T6SS harbors only a few VgrG associated effectors, the class of effectors to which DddA belongs (Cherrak et al., 2019). This is obviously less than can be achieved through heterologous expression. However, as noted in the response to point #1, our studies of the outcome of effector delivery via the T6SS to a susceptible recipient (*P. aeruginosa*) support our assertion that the heterologous expression studies we present mirror the natural intoxication process.

3) Why didn't you sequence WT genomes exposed for less time or concentration of DddA? Effect on ung background is very interesting as proof of principle but ung genotypes are absent in nature.

The reviewer raises a good point. As background, we cannot sequence highly intoxicated wild-type bacteria owing to the BER-dependent nucleoid degradation that we document in this manuscript. However, a version of the experiment described by the reviewer is feasible and was reported in our initial study describing DddA. In short, we provided low level exposure of *E. coli* to DddA (by careful immunity titration) for a series of passages that included clonal bottlenecking. Our results indicated C•G-to-T•A transitions in a 5’-TC-3’ context accumulate disproportionately under these conditions (Mok et al., 2020). We have added mention of this finding to our manuscript (subsection “DddA mediates chromosome degradation and DNA replication arrest”).

4) Please consider your findings in light of the work on the WT mutational spectrum of B. cenocepacia, which found elevated GC->TA transitions in late replicated regions: https://www.genetics.org/content/200/3/935

We thank the reviewer for pointing us to their interesting study. With respect, we question the relevance of the study to the manuscript at hand. To our knowledge, there is no evidence that T6S effectors intoxicate cells in monoculture, which are the conditions employed in the study cited. Effector-producing strains are protected by cognate immunity proteins, limiting target cells to other species or to strains of the same species that possess different effector–immunity pairs (e.g. Diniz et al., 2015, Russell et al., 2014). Additionally, the *B. cenocepacia* strain in the reviewer’s published work does not encode a homolog of DddA.

The spectrum and probability of spontaneous mutations under neutral selection in bacteria more broadly is relevant to our work and we cite such studies where appropriate. We also use one such study as the basis of our statistical analysis of DddA-catalyzed mutations in *E. coli* during interbacterial competition (subsection “Bacteria resistant to DddA-mediated intoxication accumulate DddA-catalyzed mutations”; Lee *et al.* 2012). The reviewer’s study was not considered for these purposes for several reasons: (i) a study describing neutral spontaneous *E. coli* mutations was available, (ii) the GC content of *B. cenocepacia* is significantly different from that of *E. coli*, (iii) the DNA repair pathways of *B. cenocepacia* have not been studied extensively and therefore it is unclear how they compare to those of *E. coli*, and (iv) we did not observe mutations in *B. cenocepacia* K56-2 during interbacterial competition with *B. cenocepacia* H111.

5) Subsection “Bacteria resistant to DddA-mediated intoxication accumulate DddA-catalyzed mutations”. All of the inhibitory activity due to DddA – this is an overstatement, you haven't possibly examined the range of conditions / environments by which T6SS could inhibit competitors.

In the sentence cited, we were referring specifically to the finding that a statistical difference in fitness of *B. cenocepaciadddA(E1347A)* and *B. cenocepacia∆icmF1* against *P. aeruginosa*, *P. putida*, and *B. thailandensis* was not observed in our experiments. We have revised this statement to clarify that all of the inhibitory activity of the T6SS of *B. cenocepacia* is due to DddA under the conditions of the assay (subsection “DddA intoxication consequences vary among recipient species”), which we conclude based on the observation that a T6SS-deficient strain has a similar phenotype as a DddA mutant (See Figure 3).

6) Discussion. The manuscript focuses exclusively on toxic editing of the nucleic acids of competitors, not the origins of genetic diversity. This point is speculative and not well founded by the manuscript. If you wish to make this point, please consider the native effects of overexpressing these enzymes in B. cenocepacia, or wherever the ortholog is found.

We believe the reviewer is operating under a misconception that the most likely target of DddA mutagenic activity is an organism that produces the toxin. As we note above (response #4), there is no evidence – and a wealth of direct and indirect negative data to refute – that endogenous production of the toxin leads to unchecked activity within the producing strain. Instead, our data clearly show that mutagenic activity of the toxin can occur when it is delivered via the T6SS (Figure 4). In this context, we see an increase of mutation frequency in the recipient population, and sequencing results combined with statistically analyses provide strong evidence that these mutations are attributable to DddA activity (they are enriched in C•G-to-T•A transitions in a 5’-TC-3’ context far above the rate at which this mutation arises spontaneously in the recipient species under investigation) (Figure 4B). Thus, we believe we have provided convincing evidence that DddA can exert mutagenic activity in a native context (when delivered to a contacting bacterium via the T6SS) and do not see a benefit to studying the effects of the toxin when over-expressed in the producing organism, a non-natural scenario.

7) Discussion. Again, see the article in which we actually measured mutation spectra in Burkholderia – a better reference than some cited. And further, studying mutation spectra in monocultures allows for a measure of the endogenous effects of traits like this.

The reviewer cites a study from their group that provides an assessment of the spontaneous mutation spectrum observed during monoculture of *B. cenocepacia.* However, here we are examining the mutation spectra induced by DddA delivery to *E. coli* and *K. pneumoniae,* so we are unsure of the benefit of comparing this to mutation spectra observed in *Burkholderia* (see response to point #4). Additionally, as noted in the responses to points #4 and #6, DddA is unlikely to have mutagenic activity in monocultures, as the endogenous protein is prevented from acting by expression of the native immunity determinant.

8) Discussion. About multiple mutations arising concurrently. We see no evidence of this, again see Dillon et al., 2016.

Here, we are referring to the multiple mutations we observed in a single genome following delivery of DddA to *E. coli* or *Klebsiella* during interbacterial competition. These are described in the text (subsection “Bacteria resistant to DddA-mediated intoxication accumulate DddA-catalyzed mutations”) and are documented in Supplementary file 1.

9) Discussion. The speculation about the evolution of mutators related to this phenotype is unfounded and based on a fuzzy group selection argument. Bacterial populations are rarely starved for genetic variation, and when they are, defects in DNA repair might hitchhike with beneficial alleles. There is a great deal of theory and experimentation about the conditions for the evolution of mutators – see e.g. Raynes and Sniegowski, (2014) for a good review.

We acknowledge that our argument for the potentially adaptive nature of the mutagenic capacity of DddA is quite speculative at this stage. We also acknowledge that selection for mutator alleles is often achieved via hitchhiking with beneficial mutations (e.g. Raynes and Sniegowski, 2014 as noted, Jayaraman, 2011). However, the scenario we are describing here is different from the classical problem of how mutator alleles become selected in populations. Specifically, we have shown that a toxin produced by one bacterial species can mutagenize a second species, thus transiently increasing its mutation rate. If this leads to an increase in the rate at which beneficial mutations arise in the recipient organism (as would be observed if that organism acquired a mutator allele), then we would argue that the toxin producer might indirectly benefit if the recipient organism is a cooperator.

We have revised this paragraph in our discussion in an attempt to clarify this reasoning (Discussion) and tone down our speculation.

10) Subsection “Bacterial strains and culture methods”. I do not believe any evidence is presented that this mechanism can generate diversity other than by killing competitors.

We are quite confused by this statement. Figure 3 of the original manuscript presented an abundance of evidence that DddA generates genetic diversity in resistant target cell bacterial populations such as *E. coli* and *K. pneumoniae*. These mutations were detected in the genomes of bacteria that *survived*, i.e., were not killed by, DddA.

The selection of organisms for competition experiments raises some questions. Why two E. coli strains? Why not include other Bcc organisms? In general, a broader taxonomic sampling would have been interesting and could have shed some light on ecological roles. It is e.g. tempting to assume that B. cenocepacia (a plant-associated organism turned opportunistic pathogen) experiences more competition in its natural habitat (rhizosphere) than in the lungs of CF patients. Or alternatively, its ability to kill other opportunistic pathogens may have allowed it to invade these lungs in the first place. I am not saying these questions are easy to answer, but a better global picture of DddA's spectrum of activity could be very informative.

We agree that a broader understanding of the spectrum of outcomes resulting from DddA activity across phylogenetically diverse organisms that *B. cenocepacia* is likely to encounter in different habitats would be a useful addition to the manuscript. To this end, we have added competition results and mutagenesis assays for several additional competitor species to the manuscript (see Figure 3 and Figure 4—figure supplement 1). While to date we have only observed mutagenesis of species resistant to intoxication, we acknowledge the possibility that in certain recipient species, delivery of DddA could result in killing of some cells and mutagenesis of others. Future work will be directed at assessing the impact of DddA delivery on an even broader cross-section of potential competitor species.

One criticism is that the study tackles several different objectives, each of which is only partially explored. For example, determining why some cells are resistant and others are sensitive to DddA remains and open question. What role the mutagenesis activity of DddA and its homologs may play in non-laboratory situations is currently unknown (although difficult to examine). Finally, the virulence role of SsdA1 has not yet been elucidated.

We believe this criticism is quite valid. We hope that lines of investigation within our study that were initiated and not fully interrogated will be fruitful avenues of study for our lab and others in the future.

[Editors' note: further revisions were suggested prior to acceptance, as described below.]

Essential revisions:1) It appears that DddA can increase genetic variation by roughly a log in neighboring cells. and some of these mutations could in theory be beneficial to the neighbors in some circumstances. However, it's unclear (1) how this trait would be selected and maintained in one lineage for the express benefit of unrelated others and (2) how frequently the process of mutagenizing neighboring populations would allow them to adapt beyond their unaffected capacity.

We address the reviewers’ specific concerns regarding these points in detail below.

2) On point 2: in theory, this effect would be strongest in small, clonal, structured populations as you suggest, and I agree that the unique effect of DddA might be its ability to produce multiple mutations per generation. However, I couldn't find clear evidence of multiple simultaneous mutations per cell in this study or the preceding publication. (This would likely require WGS of effectively single cells, which is tricky in this system.) Rather, results of cumulative accumulation of mutations over multiple generations of growth seem to be reported. Further, the measurement of mutation rates (e.g Figure 4) was also inferred from cumulative effects of population growth, in which jackpot effects of early-mutated cells could be amplified and lead to overestimates in mutation rates. This problem and appropriate measures are described here: https://aac.asm.org/content/52/4/1209. Please revisit your estimates of mutation rates in terms of mutational events, which require an assessment of Poisson variation among many replicates.

The reviewer raises concerns here about how we report mutation frequencies in populations and individual cells undergoing intoxication by DddA delivered by *B. cenocepacia.* It appears that we may not have provided adequate detail in the main text regarding how these experiments were conducted. Several aspects of our protocol lead us to conclude that we can reliably detect multiple mutations conferred simultaneously by DddA in a single cell, and that our mutation frequency calculations are not impacted by jackpot effects. First, DddA-mediated intoxication experiments take place on a short time scale (one hour) and are initiated with bacterial populations in stationary phase. We have confirmed there is no significant growth of the intoxicated population under these conditions. To assess mutations conferred by DddA in single cells, we plate the intoxicated populations for single colonies, and then perform whole genome sequencing on individual clones, a standard microbiological technique for assaying traits in a clonal population deriving from a single cell. From the genome sequences of clones isolated in this manner, we detect 50 SNPs with a signature of DddA activity (C•G-to-T•A transitions in a 5’-TC-3’ context) across 11 *K. pneumoniae* clones and 82 such SNPs across seven EHEC clones. Additionally, in each case, the SNPs detected were present at near 100% frequency in the population, indicating they are unlikely to have arisen during the outgrowth period. Together, these results clearly indicate that multiple mutations are installed by the activity of DddA in single cells during their brief period of exposure to the toxin (subsection “Diverse deaminase toxins have mutagenic activity”). In calculating the mutation frequencies reported in Figure 4, we similarly employed a short period of intoxication during which no detectable growth of intoxicated populations occurred, and thus jackpot effects of early mutations are avoided.

We have added a clarification to the text indicating that coculture experiments from which mutation frequencies were calculated or clones isolated for WGS were only performed for a single hour, thus limiting accumulation of mutations over multiple generations (subsection “Bacteria resistant to DddA-mediated intoxication accumulate DddA-catalyzed mutations”).

3) On point 1, this argument is featured in the Abstract, which indicates that this remains a major claim of the paper, but it's unfounded. The study presents no evidence that mutations acquired by affected cells are adaptive and increase fitness, and the conditions that might allow for evolution of the proposed altruistic effects are vanishingly rare in theory and absent in the literature. I recommend removal from the Abstract and further caution in the corresponding sentences in the Discussion.

In the Abstract of our manuscript, the adaptive benefit conferred by DddA that we were referring to is the increase in the frequency of rifampicin resistance we observe in DddA-intoxicated populations. Given the clinical importance of this antibiotic, we reasoned that acquisition of resistance can provide a fitness benefit to bacteria under certain circumstances. We have amended the Abstract to be more specific in the regards to the adaptation we are referring to. In the Discussion, we have removed one of the more speculative sentences, in acknowledgement of the reviewer’s concerns (Discussion).